# Mechanistic insights into transcription factor cooperativity and its impact on protein-phenotype interactions

Ignacio L. Ibarra[1,2], Nele M. Hollmann[1,2], Bernd Klaus [3], Sandra Augsten[1], Britta Velten [3], Janosch Hennig [1] & Judith B. Zaugg [1*]

Recent high-throughput transcription factor (TF) binding assays revealed that TF cooperativity is a widespread phenomenon. However, a global mechanistic and functional understanding of TF cooperativity is still lacking. To address this, here we introduce a statistical learning framework that provides structural insight into TF cooperativity and its functional consequences based on next generation sequencing data. We identify DNA shape as driver for cooperativity, with a particularly strong effect for Forkhead-Ets pairs. Follow-up experiments reveal a local shape preference at the Ets-DNA-Forkhead interface and decreased cooperativity upon loss of the interaction. Additionally, we discover many functional associations for cooperatively bound TFs. Examination of the link between FOXO1:ETV6 and lymphomas reveals that their joint expression levels improve patient clinical outcome stratification. Altogether, our results demonstrate that inter-family cooperative TF binding is driven by position-specific DNA readout mechanisms, which provides an additional regulatory layer for downstream biological functions.

[1] Structural and Computational Biology Unit, European Molecular Biology Laboratory, Heidelberg, Germany. [2] Faculty of Biosciences, Collaboration for Joint PhD Degree between EMBL and Heidelberg University, Heidelberg, Germany. [3] Genome Biology Unit, European Molecular Biology Laboratory, Heidelberg, Germany. *email: judith.zaugg@embl.de

Transcription factors (TFs) are essential for regulating cellular functions. This regulation is based on very specific protein–DNA interactions. To comprehend the regulation of biological processes, it is crucial to understand how TFs recognize their specific DNA binding sites[1,2].

The major determinants conferring TF binding specificity are the DNA sequence and shape readouts[3]. The former is guided by interactions between amino acids and DNA bases, while the latter is driven by a preference towards physical DNA conformations, mediated through DNA-backbone and DNA minor groove contacts. While sequence readout is a major driver of specific TF-DNA interactions, DNA-shape features improve the binding predictions of certain TF families both in vitro[4,5] and in vivo[6].

Currently, over 1600 TFs are annotated in the human genome[7]. For many of them, DNA-binding preferences, summarized as TF motifs, have been determined either through in vitro binding assays[8–12] or in vivo through chromatin immunoprecipitation followed by sequencing (ChIP-seq). However, despite the wealth of data and concurrence between in vivo and in vitro derived TF motifs[13], one of the long-standing challenges in the field is the high number of TF binding events that cannot be explained by the primary motif of the assayed TF. One proposed explanation for this phenomenon is that TFs can bind DNA cooperatively and thereby strengthen their affinity[14].

Recent studies leveraged high-throughput Systematic Evolution of Ligands by Exponential Enrichment coupled to Consecutive Affinity Purification (CAP-SELEX)[10] to identify composite sites where cooperative TF binding may occur. However, despite these experimental advances, molecular mechanisms of cooperative binding and its distinction from co-binding for most TFs remain elusive. Additionally, specific TF–TF interactions have been shown to alter sequence recognition through the formation of homodimers or heterodimers, important for driving specific biological processes[15–17]. However, we currently lack an overall understanding of the consequences behind cooperative binding. One reason for this may be the lack of appropriate computational tools to systematically interrogate cooperative TF binding and their functional associations.

To this end, here we implement a framework to determine cooperative TF binding preferences from in vitro SELEX data. We identify DNA shape as an important feature to predict cooperative TF binding, in particular for pairs between Forkhead and E26 transformation specific (Ets) members. This particular prediction is validated using nuclear magnetic resonance (NMR) spectroscopy and isothermal titration calorimetry (ITC). Through site-specific amino-acid mutagenesis, we further show that DNA shape readout likely contributes to TF-cooperativity. In vivo, these cooperative sequences are unevenly distributed across different Forkhead-Ets pairs, suggesting an additional layer of regulatory complexity. Finally, through an extensive assessment of the biological consequences of TF-cooperativity in vivo, we find that the knowledge of cooperative TF binding increases the power for discovering functions regulated by TF pairs. Specifically, for the Forkhead-Ets families, we show that a joint upregulation of FOXO1-ETV6 in chronic lymphocytic leukemia (CLL) patients was associated with significantly higher time-to-treatment values.

## Results

### Modeling TF co-binding through high-order DNA features.
In a recent study Jolma et al.[10] reported hundreds of cooperatively bound TF pairs through CAP-SELEX experiments, demonstrating that TF cooperativity is prevalent and proposing that a majority of TF pairs do not directly interact, but rather form DNA-mediated complexes. To gain more insight into the mechanisms and general rules that predict binding among the identified TF pairs, we hypothesized that features encoded in the DNA contribute to the observed interactions. To test this, we devised a framework to predict the relative affinity of TF pairs based on DNA features using CAP-SELEX data. By ranking the importance of DNA features we could then identify those that potentially drive TF co-binding.

CAP-SELEX data was obtained from Jolma et al.[10]. After reprocessing and quality control we built models to predict the relative affinity of $k$-mers (DNA sequences of length $k$) bound to TF pairs in a procedure adapted from Riley et al.[18] (see Methods section), which was previously employed to identify DNA features that determine binding of mainly single TFs[4–6,19]. Relative affinity was defined as the enrichment of a $k$-mer in the last cycle of the SELEX experiment relative to its input abundance. We compared the performance of a basic model, which predicted the relative TF affinities from the mononucleotide sequence (1mer) with models that included more complex features, such as dinucleotides, trinucleotides (2mer and 3mer) or DNA-shape (shape)[4]. 2mer/3mer/shape features have been previously proposed to capture DNA stacking interactions, local-structure elements, and the overall DNA structure, respectively (Fig. 1a)[3,4]. We implemented models using L2-regularized multiple linear regression (L2-MLR), and assessed the impact of these features on TF-binding by calculating the relative improvements measured as $R^2$ differences on testing data ($\Delta R^2$) between the full model ($1mer + shape/1mer + 2mer/1mer + 2mer + 3mer$; 12, 20, and 84 features per position, respectively) and the reduced model ($1mer$—4 features per position) using cross-validation (see Methods section).

For each TF pair, we used the reported consensus sequences[10] as references for $k$-mer selection, considering all sequences up to a defined number of mismatches, along with their relative affinity values. One challenge that arises when working with composite TF binding models is that their DNA binding regions are often very long and require high $k$ values, which leads to low coverage and hampers relative affinity estimates. To account for this, we developed a trim-and-summarize approach where we generated sets of tiled $k$-mers for each original $k$-mer of lengths no shorter than ten nucleotides, and summarized their effect on the prediction as a median $R^2$ (Fig. 1a, Supplementary Fig. 1a, b; see Methods section). This resulted in 507 composite motifs with relative affinity estimates, comprising 77 unique TFs in 280 unique TF pairs (Supplementary Data 1). We found that models including higher-order features ($1mer + 2mer$, $1mer + 2mer + 3mer$, or $1mer + shape$) performed consistently better than sequence-only models (1mer) (mean $p < 1.0 \times 10^{-6}$; Wilcoxon rank-sum test). These results indicate that higher order features are important for predicting TF co-binding (Fig. 1b, c). Notably, since the predictions were done on held-out data, the positive $\Delta R^2$ is not due to overfitting.

Overall, regardless of whether higher order sequence features are interpreted as DNA shape or as dinucleotide dependencies, our results suggest their significance in the prediction of TF co-binding.

### Forkhead-Ets co-binding is linked to DNA shape features.
We next sought to assess whether DNA shape/high-order features was important in driving co-binding between particular TF families. To do so, we compared the $\Delta R^2$ between full ($1mer + $ high-order features) and basic (1mer) models across all TF pairs stratified by family. In agreement with previous studies, we observed a moderate but significant increase in $\Delta R^2$ for TF pairs involving homeodomain members when using DNA shape[5,15,20] (Fig. 1d, median $\Delta R^2 = 0.07$; $p = 1.4 \times 10^{-3}$, one-sided Wilcoxon rank-sum test); (see Methods section). The strongest effect of

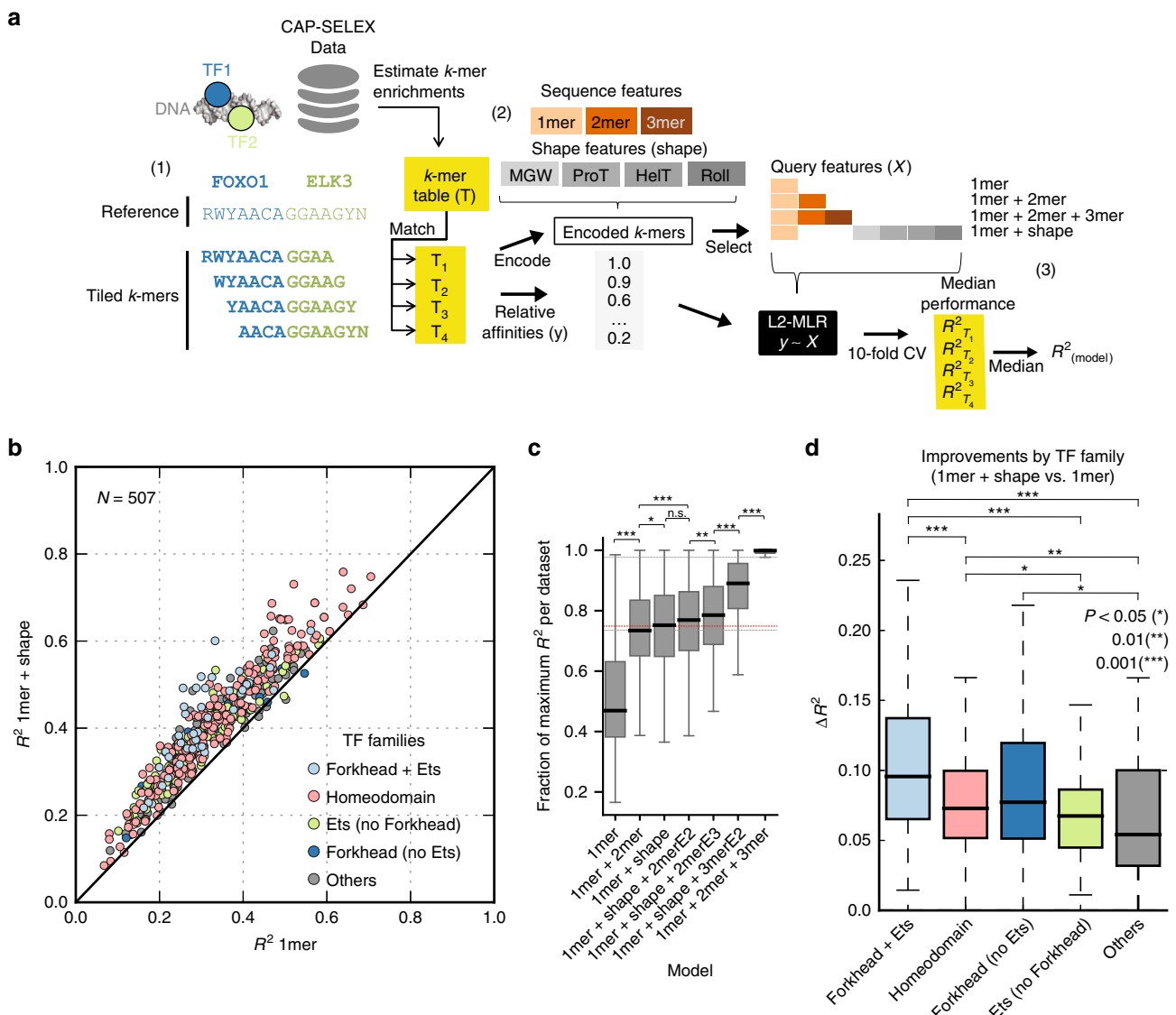

**Fig. 1 Addition of DNA-shape features improves combinatorial binding predictions in CAP-SELEX data. a** (1) Description of trim-and-summarize approach to obtain relative affinities for composite motifs (*k*-mers): a reference *k*-mer from CAP-SELEX data is trimmed from either side into multiple tiled *k*-mers with lengths no shorter than ten (blue regions assigned to consensus sequence for FOXO1 in reported *k*-mer, green regions assigned to ELK3 consensus sequence in reported *k*-mer) (2) L2-regularized multiple regression model (L2-MLR) are generated using DNA features as predictors and relative affinities as response variables (see Methods section). DNA sequence features (*1mer*, *2mer*, *3mer*) and DNA shape features (MGW, ProT, HelT, Roll), are tested in different combinations to assess their prediction contributions (3) A consensus improvement for each reference *k*-mer and model is obtained by cross-validation in each tiled *k*-mer table (10-fold CV) and calculation of the median tiled *k*-mer $R^2$ improvement for all cases. **b** Trim-and-summarize testing $R^2$ values are shown for each CAP-SELEX and reference *k*-mer combination, using tiled *k*-mers. Values above the diagonal indicate improvements in the testing set prediction performance when using mononucleotide and shape features together (*1mer + shape*, *y*-axis), with respect to models with only mononucleotide features (*1mer*, *x*-axis). Relevant TF family and TF family pairs are labeled by colors (Others = non-labeled families). **c** Fraction of maximum variance per $R^2$ dataset using different model configurations. *P*-values and boxplots defined as in **d**. **d** Trim-and-summarize $\Delta R^2$ differences between 1mer + shape versus 1mer, stratified by family (IQR = Q3 − Q1; whiskers = 1.5 × IQR) (*p* indicates Wilcoxon test *p*-adjusted values, corrected by Benjamini Hochberg's procedure[56]).

DNA shape, however, was observed for Forkhead-Ets pairs (median $\Delta R^2 = 0.09$, $p = 1.8 \times 10^{-5}$). This shape-dependency was more pronounced than the values for each family alone (Forkhead and Ets median, both $\Delta R^2 = 0.07$). No such family-specific bias was observed when using any of the other high-order models. And despite their overall $R^2$-values being higher than in the *1mer + shape* model (Fig. 1b; Supplementary Fig. 1d), the median $R^2$-values for Forkhead + Ets datasets were highest in the *1mer + shape* models (Supplementary Fig. 1e). Therefore, and since crystallographic studies have demonstrated that actual DNA

shape varies across Forkhead members[5,21] we decided to focus the analyses in the following sections on the DNA shape features, considering only the *1mer + shape* models.

Due to the known bi-specificity of Forkhead TFs[22], we performed the same analysis after discarding DNA sequences containing the alternate Forkhead motif (FHL), and obtained comparable results (Supplementary Fig. 1c). Overall, these observations highlight that DNA shape (or high-order features captured by shape) are important for predicting co-binding in a subset of TF-families, and particularly so for Forkhead-Ets pairs.

**Prediction and validation of Forkhead-Ets cooperativity**. We next wanted to gain mechanistic insights into the specific sequences that drive the co-binding between members of the Forkhead and Ets families that could potentially explain TF binding to non-canonical sites through cooperative binding. To this end, we used FOXO1 and ETS1 as a prototype Forkhead-Ets pair, and classified DNA sequences based on their level of cooperativity. Specifically, for each $k$-mer, we compared the relative affinities for ETS1 and FOXO1 obtained from their respective high-throughput SELEX (HT-SELEX) datasets (Fig. 2a), and defined their cooperativity-potential as the ratio of predicted relative affinities between FOXO1:ELK3 (ETS1 paralogue) and the mean predicted relative affinity for FOXO1 and ETS1 (see Methods section). We found that the cooperativity potential dropped with increasing FOXO1 binding affinity, while the relative affinity of ETS1 had little effect (Fig. 2b). These results indicated that the FOXO1-binding strength determines the level of cooperativity. This conclusion was further corroborated by comparing representative DNA-sequences classified as non-cooperative, cooperative and highly cooperative ($\omega$-none, $\omega$, and $\omega$-high, respectively Fig. 2c), which only differed in their Forkhead binding region. Similarly, protein-binding microarray (PBM) data[8] revealed higher affinity of Forkhead members for $\omega$-none than for $\omega$ sequences, while weak binding was observed for $\omega$-high (Supplementary Fig. 3a; see Methods section). Despite a low affinity for $\omega$-high in PBM data, we observed higher median relative affinities for this $k$-mer in CAP-SELEX datasets with both Forkhead and Ets factors than in other datasets (Supplementary Fig. 3b; see Methods section). These results and the negative correlation between Forkhead motif matches and cooperativity (Fig. 2b) suggest that Forkhead TFs can bind to $\omega$-none sequences on their own by recognizing a strong Forkhead binding site, while relying on allosteric interactions with Ets partners for recognizing $\omega$ (and possibly $\omega$-high) sequences by forming a cooperatively bound complex. Notably, crystal structure data suggest an alternate spacing between Forkhead and Ets DNA-binding domains for $\omega$-high (Supplementary Fig. 2)[23,24].

To validate the cooperativity predictions experimentally, we used isothermal titration calorimetry (ITC) to monitor changes in the dissociation constants ($K_d$) for the three DNA sequences with FOXO1 alone and in presence of ETS1. These sequences contain a strong ETS1 binding site and mainly differ in their FOXO1 binding region (Fig. 2c). For FOXO1 alone, we observed a ten-fold stronger binding for $\omega$-none than for $\omega$ ($K_d = 24 \pm 3$ nM and $352 \pm 22$ nM, respectively; $p < 0.01$, two-sided $t$-test Fig. 2d; Supplementary Fig. 4a) while no interpretable results were obtained for $\omega$-high of ITC. To test the effect of cooperativity on FOXO1 binding, we titrated FOXO1 into a mixture of each DNA sequence and ETS1 and indeed observed a significant reduction in the $K_d$ for $\omega$ ($44 \pm 11$ nM; $p < 0.01$), but not for $\omega$-none ($26 \pm 2$ nM). This indicates cooperative binding of FOXO1 and ETS1 for $\omega$ but not for $\omega$-none. As ETS1 alone binds with similarly high affinities to $\omega$ and $\omega$-high ($\omega$-none could not be measured by ITC), we propose that this cooperativity is mediated by DNA. This is also supported by the FOXO1:ETS1 co-crystal structure with DNA[25] showing no direct interactions between their DNA-binding domains (Fig. 3e).

Because ITC was inconclusive for $\omega$-high and FOXO1, we resorted to measuring NMR chemical shift perturbations (Fig. 2e; Supplementary Fig. 5), interpreted as weak, moderate, or strong binding depending on the exchange regime (fast, medium, slow) to assess cooperativity between FOXO1 and ETS1. Results for $\omega$-high were corroborated qualitatively by NMR, as chemical shift perturbations switched from fast-exchange to intermediate-exchange regimes, indicating an increase in binding affinity for $\omega$-high in presence of ETS1 (Fig. 2e, Supplementary Fig. 5). For $\omega$-none, we observed slow-exchange (stronger binding) for both FOXO1 alone and in the presence of ETS1 (Supplementary Fig. 6). For FOXO1 on $\omega$-high, we observed chemical shift perturbations in the fast exchange regime, enabling fitting and affinity determination. FOXO1 bound to $\omega$-high three orders of magnitude weaker than $\omega$-none (mean $K_d$ for selected peaks = $167 \pm 45$ μM; Fig. 2e) and changed to the intermediate exchange regime in presence of ETS1, indicating stronger binding, consistent with a cooperative interaction. The fact that we cannot measure ITC for every combination (FOXO1-$\omega$-high and ETS1-$\omega$-none) raises questions regarding the biophysical nature of these interactions, which are difficult to answer in such a multi-component system. However, we could confirm the DNA-dependent affinity changes for FOXO1 and its modulation in presence of ETS1.

To generalize our cooperativity prediction to other FOXO1-Ets pairs co-binding with equivalent conformations, we assessed the relative affinities of FOXO1 for sequences containing the binding patterns present in $\omega$-none and $\omega$ (5′-GTAAACA-3′ vs. 5′-AACAACA-3′) in single (HT-SELEX) and paired (CAP-SELEX) data. As expected, FOXO1 showed significantly higher relative affinities for $\omega$-none versus $\omega$ in HT-SELEX late rounds (Supplementary Fig. 3c, see Methods section). In CAP-SELEX, however, relative affinities for $\omega$-none and $\omega$ were similar for the majority of datasets comprising FOXO1 paired with Ets (ELF1; ELK1/3; ETV4/5), Homeodomain (HOXB13) and GCM (GCM1) members.

Altogether, our framework to predict cooperativity for FOXO1-ETS1 pairs based on combining single-TF HT-SELEX and paired-TF CAP-SELEX data was experimentally validated by ITC and NMR. Our findings suggest a widespread mechanism whereby Forkhead TFs recognize non-optimal binding sites through cooperative interaction with specific partner TFs.

**Structural insights into TF co-binding by feature analysis**. Based on the above observations, we wanted to gain further structural insights into the cooperative Forkhead-Ets interaction as well as other TF pairs. So far, we showed that DNA shape features are important to predict binding of TF pairs (Fig. 1d), and in the case of FOXO1:ETS1, differences between high-cooperative and non-cooperative DNA sequences were related to the Forkhead binding site (Fig. 2b). We therefore hypothesized that position-specific DNA shape features may determine the cooperative potential of DNA sequences. To test this, we used our modeling framework to calculate the importance of DNA shape features at each position along the composite motifs for all TF pairs in the CAP-SELEX data[10]. Specifically, we compared models with and without all DNA shape features at a given position, and reported the maximum $\Delta R^2$ per position, adapting an approach proposed by Yang et al.[5] (see Methods section). For each composite motif, we thereby obtained a shape profile that captures the importance of DNA shape at each position for predicting the relative affinities of a TF pair (Fig. 3a; Supplementary Fig. 9).

To globally explore the positional effects of shape profiles in the core motifs, we scaled binding sites across datasets to the same length and grouped them into five groups using unsupervised clustering (Fig. 3b; Supplementary Fig. 10a; see Methods section). All clusters were characterized by a single peak in the shape profile, highlighting a localized DNA shape effect along the protein-DNA interface. We observed a significant enrichment of TF pairs containing Forkhead members in cluster 1 (odds ratio = 3.1, Fisher's exact test, adjusted $p$-value < 0.1). Other TF families and TFs were enriched in other clusters

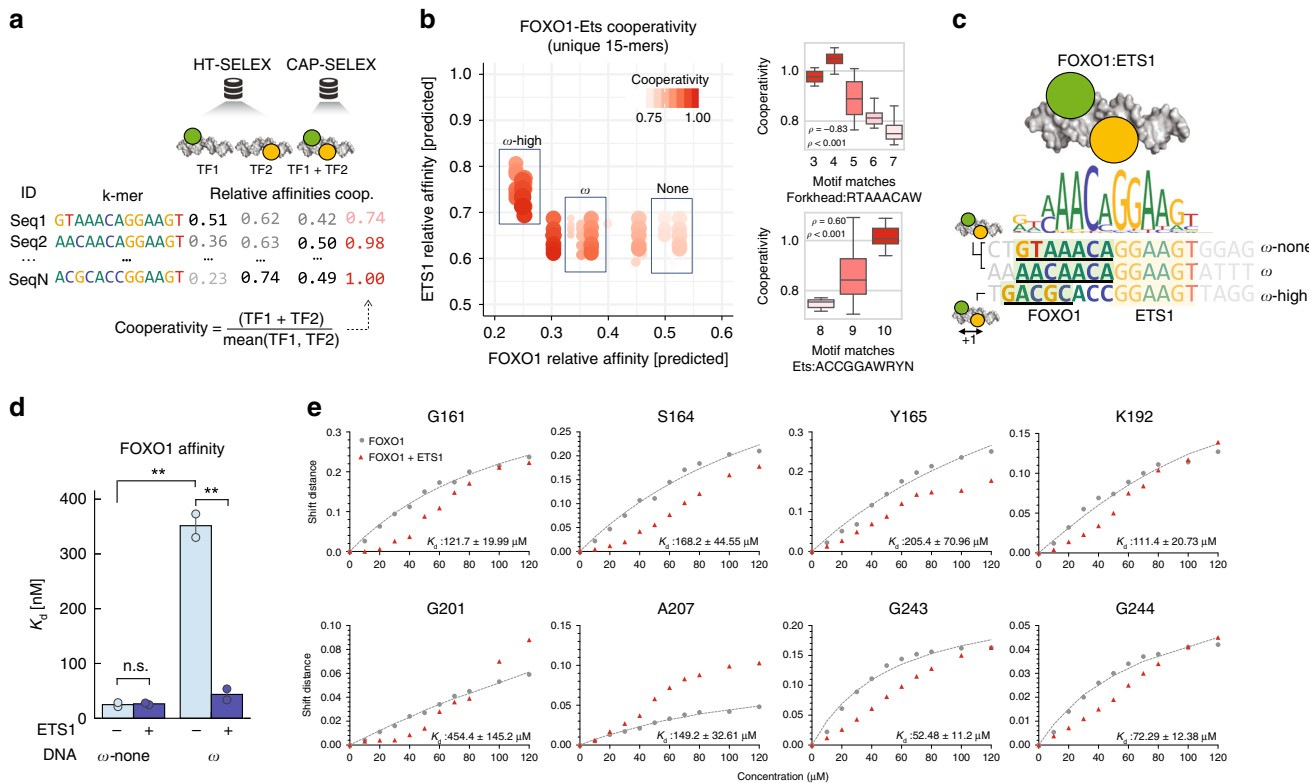

**Fig. 2 Prediction and validation of cooperative binding sites from SELEX data. a** Workflow describing the calculation of the cooperativity potential between *k*-mers from CAP-SELEX and matched HT-SELEX data. Predicted relative affinity values of *k*-mers from CAP-SELEX data are scaled by the mean value observed in HT-SELEX for the matched single TF datasets. **b** Comparison of relative affinity predictions for ETS1 and FOXO1 for 15-mers using HT-SELEX (left). *K*-mers are weighted by their estimated cooperativity potential using CAP-SELEX of FOXO1:ELK1 and HT-SELEX datasets for FOXO1 and ETS1 (ELK1 paralogue). *Y*-axis and *X*-axis indicate predicted relative affinities for ETS1 and FOXO1 datasets, respectively, using *1mer + shape* models. Correlations between number of motif matches between Forkhead or Ets consensus *k*-mer and CAP-SELEX *k*-mer in assessed region and cooperativity for Forkhead and Ets binding sites in 15-mers (right). $\rho$ and $p$ indicate Spearman's rho and significance, respectively. **c** Cartoon description of binding mode for the FOXO1-ETS1 ternary complex (top). Sequences chosen for validation from regions of none ($\omega$-none), moderate ($\omega$) and high ($\omega$-high) cooperativity are shown and aligned with Forkhead-Ets composite motif (bottom). Green and yellow highlighted regions indicate Forkhead and Ets binding regions, respectively. Underlined regions for $\omega$-none and $\omega$-none indicate Forkhead motifs, and underlined $\omega$-high region indicates a FHL site[22] with a spacing increased by one base pair (+1) between Forkhead and Ets DNA-binding domains (Supplementary Fig. 2). **d** Dissociation constant ($K_d$) ITC measurements for FOXO1 with $\omega$-none and $\omega$ DNA sequences in the absence and presence of ETS1 (ns = non-significant, \*\**t*-test *p*-value < 0.01). **e** Titration curves of FOXO1 binding to $\omega$-high using different N–H peaks, without (gray dots) and with ETS1 (red triangles). The gray line indicates the titration fit without ETS1. The different binding affinities for each residue are highlighted in the bottom-right of each panel (± = one standard deviation).

(Supplementary Fig. 10b; Supplementary Data 4). This indicates that some TF families have conserved preferences for high-order features when interacting with other TFs.

Interestingly, shape profiles for pairs including Forkhead members in cluster 1 peaked at the Forkhead binding site (Fig. 3c). Together with the results from the previous section, this suggested that shape readout at the Forkhead region might guide the cooperative interaction between Forkhead and partner TFs. This mechanism was further supported by comparing the shape profiles of exemplary Forkhead-Ets TF pairs with the profiles of single Forkhead and Ets TFs, obtained from HT-SELEX. While FOXO1 still independently showed a higher shape preference than its Ets partner, the maximum value of the profile was shifted by at least three positions relative to the one of the TF pair. Similarly discrepant patterns between single and composite profiles were observed for other Forkhead members such as FOXI1 with Ets TFs (Supplementary Fig. 10d, e). These results suggest that shape profiles are related to Forkhead-Ets cooperative binding for many members of these families, and unlikely due to individual TF binding.

**Role of ETS1 residue R409 in Forkhead-Ets cooperativity**. Given the conservation of shape profiles across many members of the Forkhead and Ets families (Fig. 3d), we next investigated whether protein-DNA interface properties at the peak of the profile may confer cooperativity. Assessing the available crystal structure for FOXO1:ETS1 bound to DNA, we observed an arginine residue (R409) of ETS1 interacting with the minor groove at the position with highest shape relevance of the FOXO1 binding site (Fig. 3e; PDB ID: 4LG0[25]). This agrees with the relevance of Minor Groove Width features for binding prediction in our models (Supplementary Figs. 7, 10c). Given the strong conservation of positively charged residues in this position across Ets family members (94%; see Methods section), we hypothesized that the DNA-cooperativity between Forkhead and Ets is mediated by this residue.

To test this, we performed site-directed mutagenesis of the ETS1-residue in question (R409) and monitored the changes in the dissociation constants of FOXO1 for one of our previously validated cooperative DNA sequences ($\omega$) using ITC (Fig. 3f). Replacement with alanine (R409A) significantly reduced binding affinity of FOXO1 to $\omega$ ($K_d = 151 \pm 11$ nM in R409A vs. $44 \pm 11$ nM in WT;

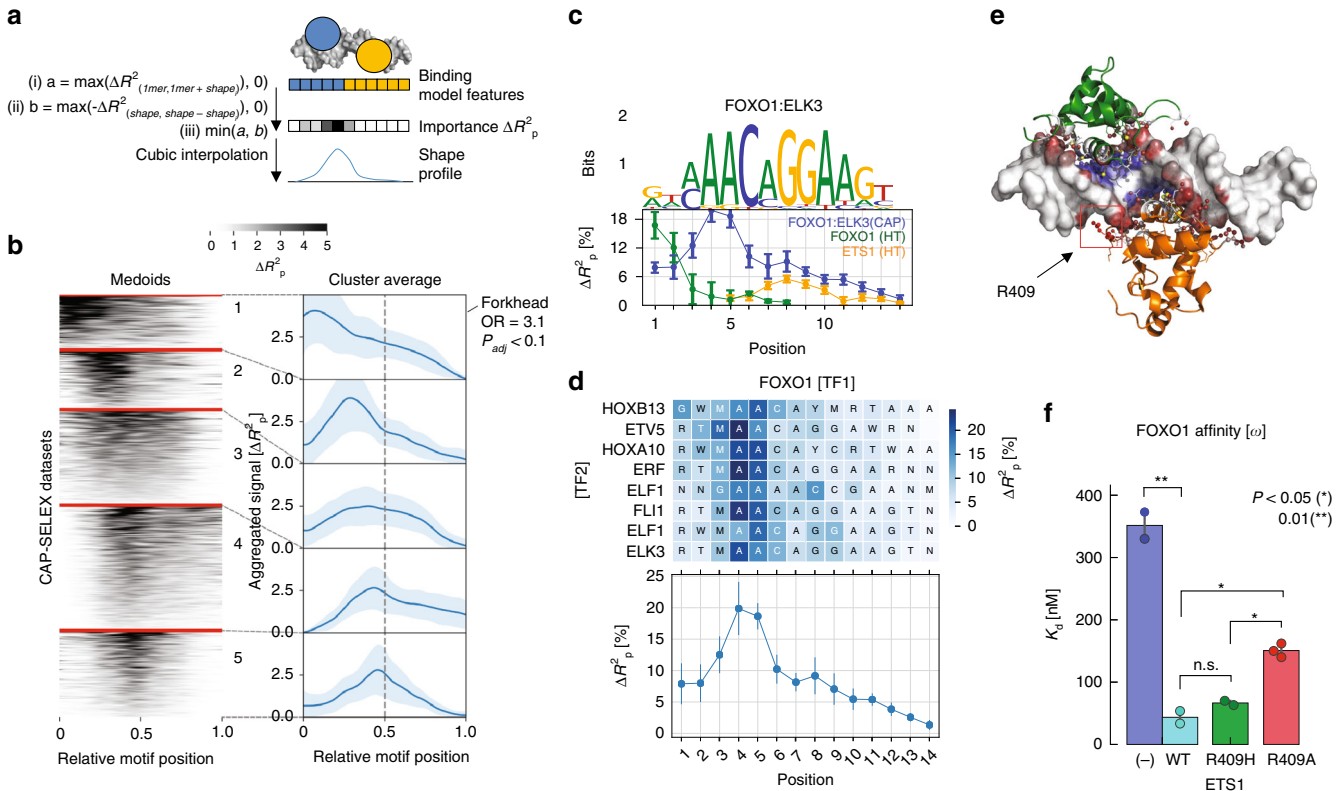

**Fig. 3 Clustering of shape improvements by position in CAP-SELEX data reveal TF co-binding shape-recognition biases. a** Scheme describing shape profiles calculation (top). For a homodimeric or heterodimeric protein-DNA complex, shape features are individually added to assess their relative contribution to the global increase in performance. Averaged contributions are transformed by interpolation to a curve representation. (Methods) **b** PAM clustering of shape profiles across all CAP-SELEX models analyzed (left). Each row shows a composite motif ($N = 438$). Five representative clusters are separated by red lines. Average shape profiles for each cluster. Blue shades indicate one standard deviation (right). Enrichments for Forkhead TFs within cluster 1 is labeled (OR = odds ratio). **c** Forkhead-Ets FOXO1:ELK3 motif (top), $\Delta R^2_p$ changes (as percentages) in FOXO1:ELK3 CAP-SELEX data (bottom) (blue line). Additional lines indicate equivalent values for FOXO1, and ETS1, an ELK3 paralogue (green and orange, respectively) calculated from HT-SELEX data[9]. Error bars indicate windowed average standard deviation, with window value of 4. **d** $\Delta R^2_p$ per position changes are shown for selected CAP-SELEX data that contain FOXO1 in combination with other binding partners of similar binding topology (top). IUPAC DNA symbols in heatmap indicate aligned k-mers. Column averages for heatmap values. Error bars indicate standard deviations in each position (bottom). **e** FOXO1-ETS1 ternary complex. ETS1 residue R409 interacting with DNA minor groove is highlighted in a red box (PDB ID: 4LG0). Visualization was enhanced using PDIViz[63]. **f** Dissociation constant measurements using ITC for FOXO1 binding to $\omega$ DNA sequence upon addition of ETS1 wild type and selected mutants (Asterisk (*) indicate adjusted p-values obtained using two sided t-test).

$p = 2.4 \times 10^{-3}$). In contrast, replacing the arginine with another positively charged residue (Histidine; R409H), resulted in a FOXO1 binding affinity similar to wild type ETS1 ($K_d = 67 \pm 4$ nM) thus retaining the cooperative interaction. To exclude that the FOXO1 affinity reduction to $\omega$ is solely driven by a decrease in ETS1 affinity upon the R409A mutation, we measured the affinity of ETS1-R409A on $\omega$ and observed no significant changes compared to ETS1 wild type (Supplementary Fig. 8a, b). We also tested a neighboring residue (Y410A), and observed weak changes in FOXO1 affinity ($K_d = 313$ nM; Supplementary Fig. 10f).

Overall, we concluded that the DNA-mediated cooperativity between FOXO1 and ETS1 is strongly linked to the interaction of R409 of ETS1 and the DNA minor groove opposite to the FOXO1 binding site. As the affinity of FOXO1 in presence of the mutant ETS1 was still higher than for FOXO1 alone, we cannot exclude that other residues contribute to the cooperativity.

**Cooperativity between Ets and Forkhead is relevant in vivo.** Having demonstrated, mechanistically analyzed, and experimentally validated cooperativity between members of the

Forkhead and Ets TF families in vitro, we next wanted to assess whether these findings could be translated to in vivo systems and whether TF–TF interactions can aid in explaining TF binding events.

We first tested whether DNA shape was equally important for predicting co-occupied ChIP-Seq regions as it was for predicting cooperative binding based on CAP-SELEX data. To do so, we used a classification framework adapted from Mathelier et al.[6], to compare models based on motif scores without (PWM) and with DNA shape features (PWM + shape) for predicting co-occupied ChIP-Seq regions between pairs of TFs (see Methods section). Overall, we obtained similar results as for the in vitro data, with 105 peak sets showing improved classification performance after addition of shape features in mapped TF composite sites ($p < 0.0001$; one-sided Wilcoxon rank-sum test) (Fig. 4a). In agreement with the in vitro data analyses, TF pairs that include a Forkhead family member particularly benefited from DNA shape ($p = 0.03$; one-sided Wilcoxon rank-sum test) (Fig. 4b).

When comparing shape profiles obtained from the in vivo and the in vitro data (see Methods section), we observed a strong agreement for 40% of them (FDR = 10%) (Fig. 4c); (median Spearman's rho = 0.25). This suggests that DNA shape plays a

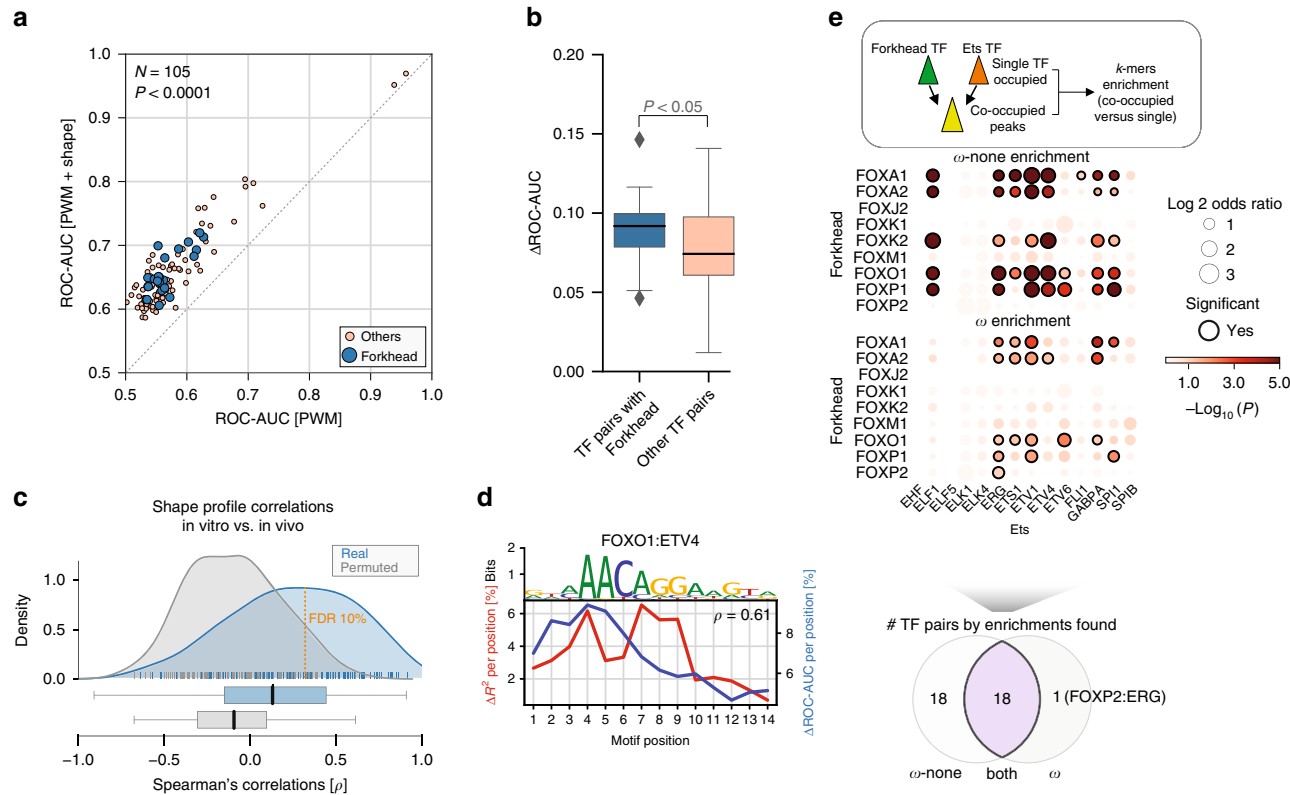

**Fig. 4 Cooperative TF binding in vivo and agreement with in vitro derived models. a** Classification performance comparison between PWM + shape (*y*-axis) vs. PWM models (*x*-axis) in regions that were selected for being co-bound by ChIP-seq for TF pairs present in CAP-SELEX data (*N* = 105). Classification performance is measured by the area under the receiver operating characteristic curve (ROC-AUC). Blue points indicate TF pairs with Forkhead as one of its members. **b** ROC-AUC differences between classification models for datasets containing at least one Forkhead member (blue) and all other TF pairs (pink). *p*-value derived from Wilcoxon rank-sum test. **c** Spearman's rho distribution of performance changes per position in in vitro ($R^2$), and in vivo (ROC-AUC) between matched TF pairs. Orange line indicates FDR 10% cutoff for positive correlations. **d** Forkhead-Ets composite motif model between FOXO1 and ETV4 (top), aligned performance changes per position observed in in vitro (CAP-SELEX; red line) and in vivo (ChIP-seq; blue line) (bottom). $\rho$ indicates effect size. **e** Scheme illustrating co-enrichment calculations for ChIP-seq regions co-occupied between Forkhead and Ets vs. single TF occupied (top). Dot plot showing $\omega$-none and $\omega$ *k*-mer enrichments between co-occupied and single TF peaks (adjusted *p*-values obtained from a Fisher's exact test between fraction of regions with motif in co-occupied peaks versus fraction of region with motif in single TF occupied peaks) (middle). Venn diagram indicating significant observation for tested *k*-mers, and number of datasets with enrichments for both (bottom).

similar role in driving cooperativity in vivo for specific TF pairs. Among the correlated profiles were several Forkhead-Ets pairs e.g., FOXO1:ETV4 (Fig. 4d).

We next wanted to assess to which extent Forkhead-Ets members bind cooperatively versus non-cooperatively in vivo. We calculated the enrichment of the $\omega$ and $\omega$-none motifs in co-occupied ChIP-Seq data for members of both Forkhead and Ets families—assuming that the FOXO1:ETS1 $\omega$ and $\omega$-none *k*-mers are for other Forkhead-Ets pairs (see Methods section). Against a background of single occupied regions, we found both $\omega$-none and $\omega$ enriched among the co-occupied regions of the 126 possible TF pairs (Fig. 4e; Supplementary Data 6). 18 TF pairs (expected by permutations = 8.9, *Z*-score = 4.0) showed enrichment for both non-cooperative as well as cooperative sequences ($\omega$-none and $\omega$) confirming the co-existence of cooperative and non-cooperative binding patterns between the same pairs of TFs in vivo. Another 19 pairs (expected = 17.4; *Z*-score = 0.9) were only enriched for either cooperative or non-cooperative sequences (1 and 18, respectively). These results suggest variable, TF-pair specific, cooperativity between the different Forkhead-Ets pairs, which adds an additional layer of regulatory complexity in vivo.

**Inferring TF-phenotype associations using TF-cooperativity.** Having shown cooperativity as both prevalent and specific

in vivo, we further investigated its potential functional impact. We wanted to assess whether certain biological processes are specifically regulated by cooperative TF binding. To do so, we assumed that genes regulated by TF pairs should reflect biological functions common to both TFs and that these functions should be captured by gene ontology terms. Further, we defined TF-pair-to-gene links by mapping regions co-occupied by both TFs (using ChIP-Seq from Remap[26]) to target genes (using GREAT[27]; see Methods section).

With this, we devised an ontology association probability that quantifies relationships between TF pairs and ontology terms using logistic regression. For each TF pair we modeled their membership to a given ontology term based on the number of their target genes and regulatory elements (normalized as *Z*-scores) (Fig. 5a; see Methods section). To test the effect of cooperativity on the ontology association probability, we compared models using (i) only ChIP-Seq data (*peaks*), (ii) only cooperativity *k*-mers (*k-mers*), or (iii) both (*peaks + k-mers*) as input. For all models, we observed higher association probabilities for ontology terms annotated to the TF pair (*TF1 and TF2*) than for random background terms (effect size[28] *r* = 0.3, *p* < 0.001; Wilcoxon one-sided test) (Fig. 5b). The highest associations were obtained for models considering genes regulated by cooperatively bound peaks and *k*-mers (mean *r* between *peaks + k-mers* vs. *peaks* or *k-mers* = 0.1; mean

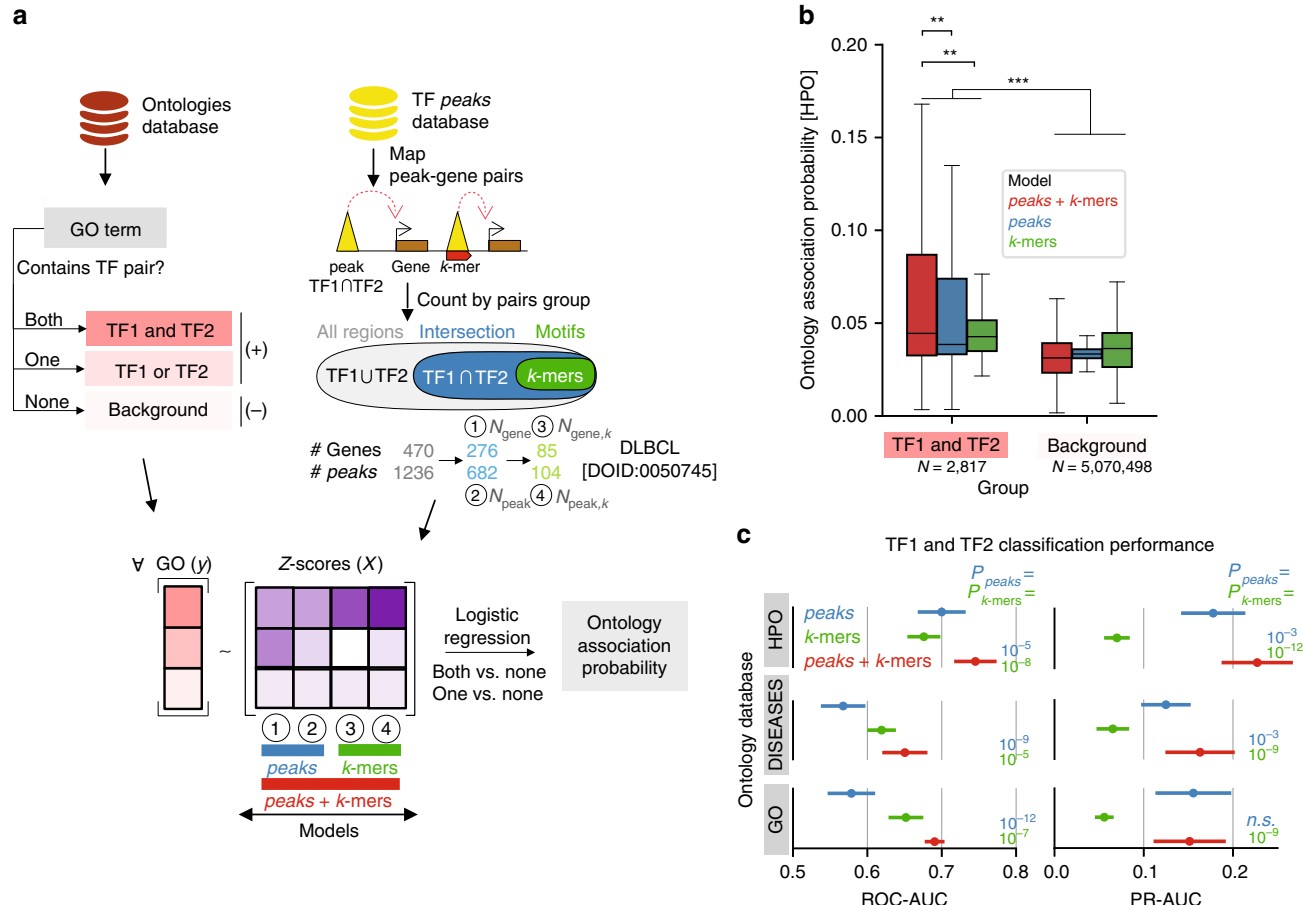

**Fig. 5 Inference of TF-phenotype associations using TF-cooperative k-mers. a** Scheme illustrating calculation of Ontology Association Probability values using TF-TF k-mers, ChIP-seq data and ontology databases. For each TF-pair and ontology combination, four metrics describing the numbers of genes and peaks proximal to the ontology-related genes are converted into Z-scores (see Methods section). From peaks assigned to for both TFs (TF1∪TF2, gray oval), the sub-selection using the co-occupied peaks (TF1 ∩ TF2, blue oval; features 1 and 2) allows calculating $N_{gene}$ and $N_{peak}$, and co-occupied peaks with TF-TF k-mers allow calculating $N_{gene,k}$ and $N_{peak,k}$ (green oval; Features 3 and 4). Ontologies are labeled by the presence of both TFs (TF1 and TF2), only one (TF1 or TF2), or none (background) in the ontology. Models with different combinations of features are tested (peaks = 1 and 2 (blue); k-mers = 3 and 4 (green); peaks + k-mers = 1, 2, 3, and 4. (red)) (DLBCL = Diffuse large B-cell lymphoma. In example, genes are mapped from DOID:0050745 in DISEASES database[32]). **b** Distributions of association probabilities for Human Phenotypes Ontology (HPO) terms for terms labeled as TF and TF2 vs. background terms are shown (Asterisk (*) indicate Wilcoxon rank-sum test p-values). **c** Classification task performances in the assessment of TF1 and TF2 vs. background terms in three ontology databases. ROC-AUC and PR-AUC indicate median areas under the ROC and Precision-Recall curves. Lines indicate 1.5 standard deviations above and below the median value. p-values are derived from 10-fold cross validation metrics comparisons between peaks + k-mers and color-matched approaches using an independent t-test, after Benjamini-Hochberg correction[56].

$p < 0.01$), emphasizing the role of TF cooperativity in regulating specific processes.

We next used the derived metric for discovering biological functions of cooperatively bound TFs. Specifically, we used the association probabilities to predict ontology terms of all TF pair combinations (see Methods section). Using defined ontology terms common to both TFs as a gold standard, performance metrics indicated that predictions were better when using peaks + k-mers vs. peaks or k-mers alone. This was the case in three tested ontology databases (GO/DISEASES/HPO)[29–32] and irrespective of the performance metric (mean ROC-AUC = 0.70 (peaks + k-mers), 0.61 (peaks), and 0.64 (k-mers) ($p = 6.7 \times 10^{-8}$); mean area under the precision-recall curve (PR-AUC)) = 0.18, 0.15, and 0.06 ($p = 2.3 \times 10^{-9}$)) (Fig. 5c). Interestingly, the classification performance of ontologies related to both TFs is higher than the one where only one TF of the pair is associated to the ontology (TF1 or TF2; mean ROC-AUC = 0.66; mean PR-AUC = 0.14; Supplementary Fig. 11a). These results indicate that processes

cooperatively regulated by two TFs can be distinguished from those regulated by each TF individually.

To capture the strongest associations between TF pairs and terms across all tested ontology databases, we defined a model-dependent, signal-to-noise threshold on the association probabilities. Briefly, we defined strong and weak associations between TF pairs and ontologies by shuffling the labels that describe whether a TF pair is part of the listed genes in the ontology (predicted variable) and recalculating the association probabilities that these decoy associations would generate when fitting a model to them (see Methods section); this recovered 6600 strong associations with high probabilities and both TFs as members of the ontology (Supplementary Fig. 5b; Supplementary Data 7). We consider this number an underestimate limited by the availability of ChIP-Seq data. This is supported by a variation of our model using only k-mers nearby transcription start sites (TSS) and no ChIP-seq data, which identified cooperative TFs linked to specific cell differentiation and disease (Supplementary Fig. 11c).

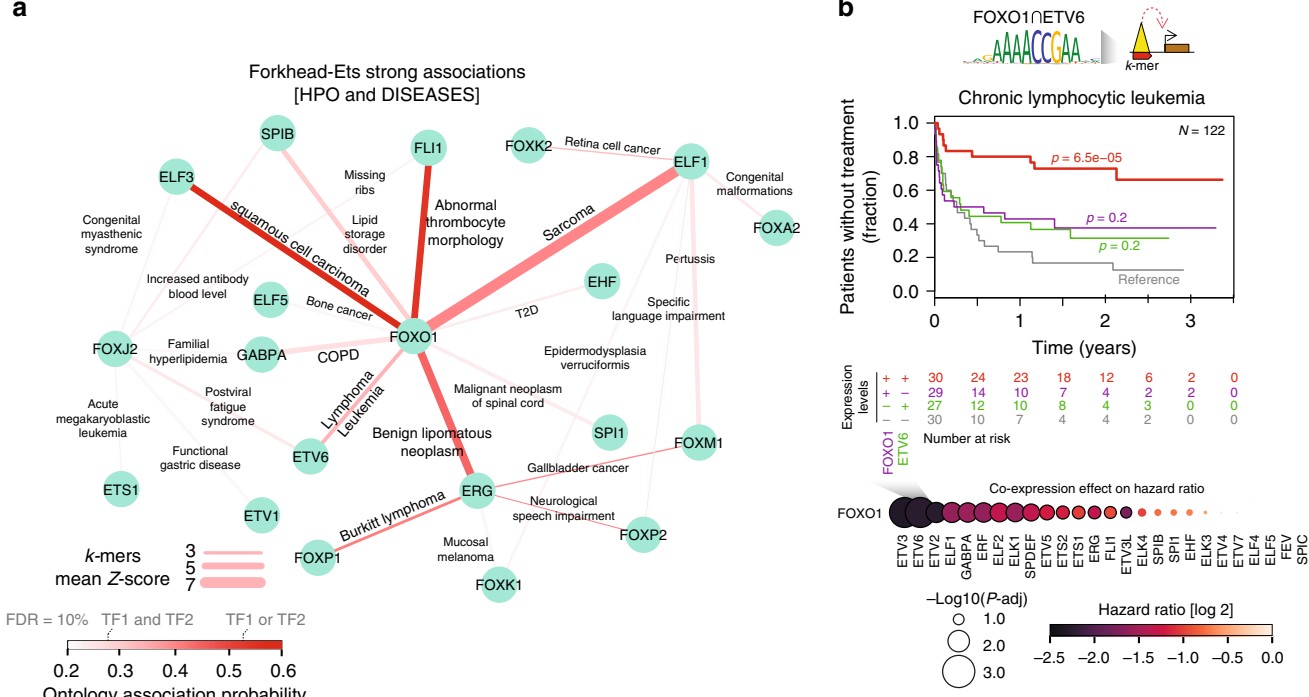

**Fig. 6 Functional and disease processes associated to cooperative Forkhead-Ets binding. a** Network of strong associations terms between Forkhead and Ets families in HPO and DISEASES ontologies (see Methods section). Nodes indicate TFs and edges indicate ontologies names. Edge width indicates mean of $k$-mer based $Z$-scores (features 3 and 4; see Methods section) (COPD = Chronic obstructive pulmonary disease; T2D = Type 2 Diabetes). FDR thresholds in color scale indicate *TF1 and TF2* and *TF1 or TF2* confidence for association probabilities obtained using the HPO ontologies database (extended thresholds in Supplementary Data 7). **b** Cartoon describing association between Forkhead + Ets $k$-mer GAAAACCGAANM and lymphoma associated genes through intersected peaks (top). Kaplan-Meier plot of time-to-treatment in Chronic lymphocytic leukemia patients when using FOXO1 and ETV6 expression medians (middle) (high and low expression are defined as above and below median values, and labeled as + and −, respectively). *p*-values are derived from two-sided log-rank comparison with respect to −/− expression levels for both FOXO1 and ETV6 (data from Dietrich et al.[36]). Dot plot showing co-expression effect on time-to-treatment for all pairs between FOXO1 and members of the Ets family (bottom). Values are sorted by decreasing significance (from left to right).

Following up on Forkhead-Ets pairs we recovered strong associations between 20 TF pairs and specific ontology terms (Fig. 6a). FOXO1 showed the highest number of associations with different TFs ($N = 9$), suggesting a multifunctional role for this TF through cooperative TF partner interactions. FOXO1 was most strongly associated with sarcoma (DOID:1115, with ELF1), and squamous cell carcinoma (DOID:5520, with ELF3) ($k$-mer = WAAACAGGAAG for both terms; mean $k$-mers $Z$-score > 5). This is in agreement with previous reports proposing FOXO1 as a prognostic marker in sarcomas[33], ELF3 in squamous cell carcinoma[34] and the recognized role of ELFs in sarcomas[35].

In light of these results and their strong literature support, we hypothesized that expression levels of FOXO1, together with predicted TF partners could be a potential readout to interrogate clinical cancer data. We examined this concept using available data on lymphoid leukemia patients to examine the effect of predicted associations with cooperative binding of FOXO1 and ETV6 (DOID:0050745, $k$-mer GAAAACCGAANM; mean $k$-mers $Z$-score = 3.2). Specifically, we stratified patients in a chronic lymphocytic lymphomas (CLL) cohort[36] into high/low expression levels for both TFs (see Methods section), to explore their usage as prognostic markers. Strikingly, we obtained a significant increase in time-to-treatment when both TFs were highly expressed (Hazard ratio (HR) = 0.21, 95% CI 0.10–0.45; $p = 6.5 \times 10^{-5}$) (Fig. 6b). This association was not found when considering each factor separately, and it was not confounded by p53 and IGHV mutation statuses (HR = 0.19, 95% CI 0.07–0.48, $p = 5.0 \times 10^{-4}$, Supplementary Fig. 11d). Importantly, this is the

strongest association to time-to-treatment among all FOXO1-Ets combinations, for which ChIP-seq data was available. Together with reports of FOXO1 and ETV6 as putative tumor suppressors in lymphomas[37,38] this suggests an important role of this TF pair in lymphoid leukemia.

Overall, our results demonstrate that cooperative TF-binding models applied to in vivo data provide a powerful tool for unbiasedly identifying the function and disease implication of TF pairs.

## Discussion

Here we provide a framework to study different types of TF binding data for single and co-binding TFs both in vitro and in vivo. This study enables us to systematically gain structural insight into TF cooperative binding, and reveal their functional and disease-related relevance. Statistical learning proved to be an integral part to understand the contributions of DNA features to TF co-binding, such as in approximating positional relevance of nucleotide interactions and DNA-shape features in TF binding models. Thereby our models provide a platform for generating hypotheses regarding the potential consequences of disruptions in either base or shape readout in cooperative TF binding[5,15,19,39]. Importantly, applying those concepts to cooperative TF binding data, we inferred specific and conserved binding preferences across TF families. Using FOXO1 and ETS1 as representative members of the Forkhead and Ets families, we demonstrate that such conserved TF-interactions are linked to DNA-shape readout with stronger effect sizes than the ones for homeodomain pairs[15].

Although these results cannot be generalized to all TFs and are limited to Forkhead:Ets pairs, they highlight the relevance of our modeling for detecting such specific interactions in the core motif, which we also observed in the CAP-SELEX data for other pairs, yet with weaker enrichments. We reinforce this argument by identifying a conserved residue that mediates cooperativity in Ets family members. As this residue happens to harbor multiple DNA-binding domain polymorphisms[40], the extent of this particular cooperativity between Forkhead-Ets members can be prone to variation across healthy individuals.

Our work presents a major methodological advance over recent studies on the quantitative assessment of DNA-shape readout and its contribution to TF binding, which are limited by data sparsity due to long binding (composite) motifs. To estimate feature preferences for such motifs, we introduced a trim-and-summarize approach, allowing the reliable quantification and comparison between models considering motifs spanning a mean of 18 base pairs in CAP-SELEX data from Jolma et al.[10] (see Methods section; Supplementary Data 1). Despite data sparsity impeding the generation of models including features beyond the core motifs[41], the reasonable agreement between our results and in vivo data, suggests this approach could prove useful in integrating low-coverage SELEX data with other studies of higher data quality[42,43], as well as screening for cooperative TF binding sites in vivo.

TF binding has been associated to chromatin regulation[44] and disease[45], yet cooperative binding has not been systematically analyzed in such contexts. The knowledge of TF–TF allostery can be used to predict co-occupied TF regions and annotate cryptic binding sites[46]. As genetic disruptions in such TF-cooperativity regions are important to understand failures in developmental programs[15] and disease[47], there is a requirement for models that predict TF binding preferences when acting in specific combinations and the functional consequences of such events. Here, the integration of cooperative TF $k$-mers with ontology associations of TFs allowed us to thoroughly examine potential functional consequences stemming from genome loci targeted by cooperative TF-binding. Although other studies have associated composite motifs to specific cell types using in vivo data before[48,49], we successfully demonstrated that incorporating a layer of knowledge on the degree of cooperative binding gives a significant leverage in identifying biological processes specific to TF pair binding. In fact, the knowledge of cooperative $k$-mers jointly used with ChIP-seq data translates into better TF-ontology predictions and could thus increase the extent of our functional knowledge on cooperative TF binding and its underlying biology (Fig. 7). Importantly, we release our current predictions for community examination of cooperative interactions between TF pairs.

Systematically understanding the interplay of TFs in disease has great potential to translate into clinically relevant findings. Our investigation of the TF pair FOXO1:ETV6 and its cooperativity-driven association with clinical features in CLL exemplifies this clearly and is reinforced by the observation of a FOXO1:ETV6 gene fusion in leukemia patients[50]. Both TFs have also been described as putative tumor suppressors in lymphomas[37,38], yet the extent of the cooperativity-driven functional impact on leukemia relative to other FOXO1-Ets combinations has not been understood nor quantified. Future work will be required to understand whether this particular mechanistic relationship occurs prior or after FOXO1 mutations[51] or whether it represents an independent event in cancer progression. Modeling of such associations and their network interdependencies remains, however, an indispensable component in leveraging TF cooperativity for functional interrogation and prioritization of disease-related TF combinations.

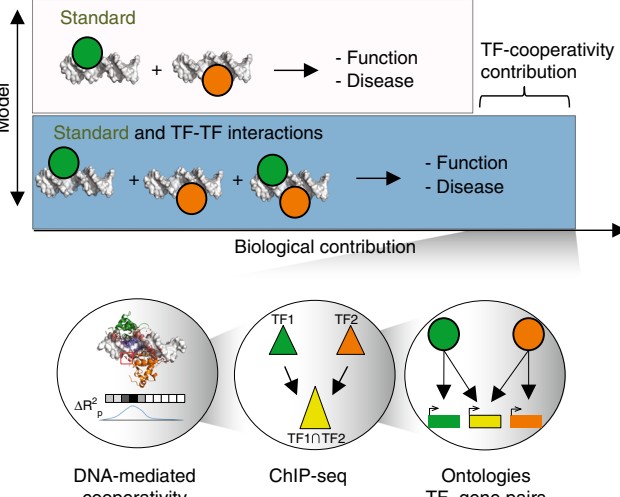

**Fig. 7 Model to estimate cooperative TF-binding biological contribution to TF-ontology associations.** Illustration of different types of models that describe associations of TF with function and disease (top). The Standard model describes the biological contribution of single TF-DNA binding, which can be improved by the addition of TF–TF interactions for an enhanced understanding of function and disease. This is translated into an overall increased discovery of strong phenotypes associated to TFs when acting in combination. Information used in this work to describe TF cooperativity and reveal TF-cooperativity linked processes (bottom) (DNA-mediated cooperativity = TF–TF $k$-mers and prioritization of important binding modes; ChIP-seq = co-occupied peaks for TF pairs; ontologies and TF-gene pairs = associations between co-occupied regions by TF pairs and their associated genes, linked to function through ontology data).

## Methods

**Transcription factor binding datasets.** Transcription factor (TF) binding data analyzed in this work were collected from in vitro and in vivo studies. Specifically, CAP-SELEX and HT-SELEX sequencing reads were retrieved from the European Nucleotide Archive, under accession entries PRJEB7934, PRJEB7934, and PRJEB20112. protein binding microarray (PBM) data were downloaded from the UniProbe database[52]. ChIP-seq peak datasets were collected from the ReMap2 database[26].

**In vitro data preparation.** The first step in the computational processing of SELEX data is the generation of count tables for $k$-mers (sequence patterns of length $k$), for each experiment where a TF or a TF pair was probed. CAP-SELEX sequencing data used to generate count tables always comes from a fixed selection round (positive) and is compared against an input library (background, or round zero). For each TF pair, we select the positive round where a binding motif targeted by the two TFs is overrepresented in the reads for a given topology vs. all other possible topologies[10]. From this, the initial value of $k$ is the length of the reported reference $k$-mers, trimming out ambiguous nucleotides in the flankings (IUPAC=N). For example, the reference $k$-mer GAAAACCGAANM has a length of 12, and thus $k = 12$. If more than one reference $k$-mer is enriched in one dataset, those are processed independently.

Once $k$-mer tables are defined, relative affinity estimates for each $k$-mer are obtained from the counts of each $k$-mer observed in the positive round, versus the amount estimated in the input data (round zero) using a fifth-order Markov Model. This correction takes into account sequencing biases[18]. Given this information, for a $k$-mer $k$ in selection round $r$, its relative affinity $S(k,r)$ is calculated as

$$S(k, r) = {}^{1+r}\sqrt{\frac{P_{\text{obs}}(k, r)}{P_{\text{exp}}(k, r)}},$$

where $P_{\text{obs}}(k, r)$ is the fraction of counts for $k$ in $r$, and $P_{\text{exp}}(k, r)$ is the expected fraction of counts for $k$ in round $r$. The derivation of the formula has been extensively described in previous work[18].

From the $k$-mer tables and their relative affinity estimates, we further subset this table for $k$-mers with high similarity between those and the reference $k$-mers indicated to be enriched, allowing up to $m$ mismatches[5]. The $m$-value threshold is proportional to the consensus sequence length and the information content for

each of its nucleotides, using the proposed formula by Yang et al.[5]:

$$m = \left\lfloor \frac{L-4}{2} \right\rfloor + 1,$$

where $L$ is the length of the consensus sequence, corrected by the ambiguity of each nucleotide. For example, GAGCA has an $L$-value of 5, but RRGCA has an $L$-value of 4, as R (purine) can represent either G or A. Datasets and reference $k$-mers used are listed in Supp. Data 1.

**Tiled $k$-mer tables**. There is an exponential decrease in the counts recovered per $k$-mer and the value of $k$, which prevents the calculation of robust $k$-mer tables and robust relative affinity estimates for high $k$ values. To overcome this, we trimmed nucleotides from both flanking regions of each consensus sequence in the list derived from the CAP-SELEX data. We thereby obtain tiled $k$-mer tables with sufficient counts for further analyses. To avoid lower complexity of DNA sequences, tiled $k$-mer tables with a length lower than ten are not considered for further analyses. Our trimming approach was benchmarked through a comparison between the effect of shorter $k$-mers on the final performance metrics (see Trim-and-summarize coefficient of determination section and Supplementary Figs. 1a-b and 8). To avoid relative affinity estimates with low support, thresholds of counts per $k$-mer are defined[5,18]. In this work, $k$-mers derived from CAP-SELEX data are discarded if the number of counts supporting those are lower than 20 counts.

**Regression models to model $k$-mer relative affinities**. To relate binding affinities with sequence and/or shape features, we used L2-regularized multiple linear regression (L2-MLR), with the following formula:

$$y = \beta_0 + \beta_1 X_1 + \beta_2 X_2 + \ldots + \beta_n X_n,$$

where $y$ is the vector of relative affinities for each $k$-mer in the $k$-mer table, $X$ ($i = 1, \ldots, n$) represent a concatenated set of features that encode their respective DNA sequences, $\beta_i$ ($i = 1, \ldots, n$) represent the regression coefficients, and $\beta_0$ represents the intercept. To prevent overfitting, L2-regularization employs an additional penalty term on the coefficients in the loss function $L(\beta)$, i.e., coefficients are obtained by minimizing

$$L(\beta) = \sum_{i=1}^{n}\left(y_i - \sum_{j=1}^{p}\beta_j X_{ij}\right)^2 + \lambda \sum_{j=1}^{p}\beta_j^2$$

with $\lambda$ set to 1.

For regression models based on DNA sequence features, the baseline models are named *1mer* and are defined by mononucleotide representations of each $k$-mer. At any $k$-mer position $i$, four features $4i$ to $4i + j$ with $j < 4$ are defined based on the nucleotide identity of $k_i$:

$$X_{N_j, 4_{i+j}} \begin{cases} 1, & \text{if } k_i = N_j \\ 0, & \text{if } k_i \neq N_j \end{cases}$$

with $N_0 = A$, $N_1 = C$, $N_2 = G$, $N_3 = T$.

In total, *1mer* models require $4k$ features for each sequence of length $k$ to fully encode its sequence in numbers. For *2mer* or *3mer* models, dinucleotide or trinucleotides are also converted into coefficients, thus requiring more features per position. For *2mer* model features, 16 coefficients between $16i$ to $16i + j$ with $j < 15$ features are necessary to describe the dinucleotide identity of each $k$-mer position and its immediate right-nucleotide.

$$X_{N_j, 16_{i+j}} \begin{cases} 1, & \text{if } k_i k_{i+1} = N_j \\ 0, & \text{if } k_i k_{i+1} \neq N_j \end{cases}$$

with $N_0 = AA$, $N_1 = AC$, $N_2 = AC$, $N_3 = AG$, $\ldots$, $N_{15} = TT$.

Similarly, for *3mer* models 64 features representing all the possibilities for trinucleotides are required. In general, for an $N$-mer model where $N \in Z^+$, $4^N$ would be required per $k$-mer position. Combinations of these models require the sum of features for each individual model, per position. For example, *1mer* + *2mer* models require $4^1 + 4^2 = 20$ coefficients per position. Equivalences between some of these models are further described by Yang et al.[5]

Models that include DNA-shape features are labeled with the keyword *shape* (e.g., *1mer* + *shape*), and consider DNA structure estimated for each tested DNA-sequence in all datasets, defined as descriptors of the overall DNA structure for that sequence. These values are listed in a DNA pentamer table, and are obtained from the DNAShapeR package[53] centering each feature value on the middle nucleotide of the pentamer. In this work, we considered the four main features provided in the original version of this table: Propeller twist (ProT), roll, helical twist (HelT), and minor groove width (MGW). In addition to these values, second order shape values are obtained by calculating the product of features in two consecutive positions, as a way to describe longer structure features. For that reason, four main shape and 4 second order shape features are required per position, allowing for eight features per position to be described in shape models. Additively, *1mer* + *shape* models require $4 + 8 = 12$ features per position where a centered DNA pentamer exists.

Flanking positions cannot be described by shape features, as these miss one or two nucleotides to successfully map a DNA pentamer. Solutions such as describing

the flanks as *3mer* features have been proposed (*1mer* + *3merE2*, where *E2* represents *3mer* features on the two end positions)[5]. In this work, we extended the shape model features to include flanking regions as well by including the average feature value of all pentamers that contain a common tetramer or trimer as found in the flanking region. Briefly, whenever a shape feature in the flanking regions is required, we average pentamer shape features that contain a fixed trimer (16 options) or tetramer (4 options). This is done with similar rules and upstream or downstream of the $k$-mer flank, according to the 5′ to 3′ directionality (left flank = upstream trimming, right flank = downstream trimming), respectively. We calculated errors for each DNA-pentamer to estimate the amount of uncertainty for each calculation using all trimers and tetramers available in the dataset in comparison with all DNA-pentamers (Supplementary Data 2). Shape features based on averaging across trimers and tetramers are closer to real pentamer DNA-shape features than the global mean generated by using all 1024 DNA pentamers or scrambled versions. In this work, we refer to shape features as models that include these flanking features.

**Trim-and-summarize coefficient of determination**. For each tiled $k$-mer table in each dataset, we use a 10-fold cross validation scheme to randomly separate the table into 10 fixed groups of equal size, iteratively fitting L2-MLR models with nine out of ten groups, and then assessing coefficient of determination ($R^2$) in the held-out group. This is done using scikit-learn[54]. As a summary statistic for each tiled $k$-mer table, we report the median $R^2$ of all held-out groups (Supplementary Data 1 and 4). As a quality control and to remove datasets with low variability and enrichment for mapped $k$-mers, at this stage we filter out datasets whose minimum testing $R^2$ value across all models for all tiled $k$-mer tables is lower than zero (i.e., model is worse than using the mean of all values as a single feature).

To generate a global $R^2$ for each dataset and model combination, we calculate the median of all median 10-fold CV $R^2$ values in each tiled $k$-mer table when using a reference $k$-mer. We refer to this as the "trim-and-summarize" $R^2$ performance, and use this number for global performance comparisons across models and datasets (Fig. 1a; Supplementary Data 1). To validate that this metric is a robust approach to obtain global $R^2$ values without major information loss, we tested whether this approach provides similar $R^2$ statistics to the ones reported by Yang et al. in HT-SELEX data[5], comparing reference $k$-mers and tiled $k$-mers. Globally, $R^2$ statistics are in strong agreement, defined as a difference of less than three nucleotides between models for reference and tiled $k$-mers. Hence we conclude that $R^2$ values obtained through trim-and-summarize are indicative of longer $k$-mer $R^2$ values as long as the length difference between initial $k$-mers and tiled $k$-mers is three or less (Supplementary Fig. 1b).

**TF family improvements in co-binding predictions**. We assessed the relationship between our trim-and-summarize $R^2$ improvements by DNA-shape features and specific TF family membership of each studied pair of TFs. Briefly, each annotation for a TF to a particular protein structure family is retrieved by the JASPAR database[55]. Significant increases in $R^2$ are assessed using a Wilcoxon rank sum test (wilcox.test in R), with $p$-values being corrected using a Benjamini Hochberg procedure (p.adjust in R)[56].

To discard bispecificity in the Forkhead + Ets datasets as a feature explaining the $R^2$ improvements, we repeated the calculation for these datasets discarding all $k$-mers containing the pattern GACGC up to one mismatch (Supplementary Fig. 1d).

**Cooperativity estimations in matched SELEX data**. To estimate TF cooperativity, we used relative affinities obtained from CAP-SELEX and HT-SELEX data and their predicted scores from *1mer* + *shape* models. We define the ratio between predicted relative affinities for a TF pair and matched single TF datasets, to estimate how close a CAP-SELEX score is to the average score in matched HT-SELEX data that would be expected for non-cooperative binding. Hence, the cooperativity for a $k$-mer $k$ is defined as

$$\text{Cooperativity} = \frac{S_{ab}(k)}{\text{Mean}(S_a(k_a), S_b(k_b))},$$

where $S_{ab}(k)$ is the predicted relative affinity in CAP-SELEX for TFs $a$ and $b$, and $S_a(k_a)$ and $S_b(k_b)$ are the predicted relative affinity estimates obtained for TFs $a$ and $b$ in HT-SELEX data for subsequences $k_a$ and $k_b$ that are contained in $k$. Lengths for $k_a$ and $k_b$ are selected based on work by Yang et al.[5] This score can be used to calculate the relative cooperativity for specific DNA sequences within a TF pair given the three experiments that are available. In this study, we limited calculations to DNA sequences that contained motifs associated to at least one TF. This was done to prevent calculating cooperativity estimates for DNA sequences linked to amplification and sequencing biases, on the contrary to TF binding specificity.

Since HT-SELEX data was for ETS1, we used the FOXO1:ELK3 CAP-SELEX dataset to estimate cooperativity factors for the TF pair FOXO1:ETS1, as it contains one common member, FOXO1, and ELK3 is a paralog to ETS1. To generate $k$-mer tables, we used $k = 13$ for FOXO1:ELK3 (reference $k$-mer: RTMAACAGGAAGT), $k = 12$ for ETS1 (NNNNGGAANNNN), and $k = 8$ for FOXO1 (RTAAACAW). This setup allows us to measure cooperativity estimates using *1mer* + *shape* models

that contain the Forkhead-Ets 13-mer binding pattern, plus two 3′ flanking positions to align the ETS1 binding model (minimum $k = 15$).

**Analysis of Forkhead + Ets sites using PBM and CAP-SELEX**. To examine the DNA binding affinity of Forkhead TFs for ω-none, ω, and ω-high, we used PBM data from the UniProbe database and compared E-scores for all 8-mers containing the patterns GTAAACA, AACAACA, and ACGCACC across all available Forkhead family members. The E-score threshold of 0.35 was used to define high-affinity sites. For CAP-SELEX datasets, 13-mers connected to those three sequences were compared by averaging the relative affinities of all 10-mers contained within those into a single value.

**Protein cloning, expression, and purification**. The ETS1 (331–440) and FOXO1 (143–270) sequences were purchased using Geneart (ThermoFisher). These were amplified and cloned using restriction-free cloning into a pETM-22 vector, which comprises a cleavable N-terminal His$_6$- and Trx-tag. The single mutations were inserted in the pETM-22-ETS1 (331–440) vector using site-directed mutagenesis.

Both proteins were expressed and purified from *E. coli* BL21 (DE3), grown in LB medium. The cultures were induced with 0.2 mM IPTG at an OD$_{600}$ of 0.8 and grown overnight at 18 °C.

After resuspension of the cells in a buffer containing 50 mM Tris (pH 7.5), 300 mM NaCl, 0.5 mg/ml lysozyme, EDTA free protease inhibitor (Roche) and Benzonase, the cells were lysed using a French press. The cleared lysate was applied to a first Ni-NTA column and afterwards, eluted with an imidazole gradient from 0 to 300 mM. The eluted protein fractions were then cleaved with 3C-protease overnight at 4 °C to remove the His$_6$-tag and simultaneously dialyzed against 0 mM imidazole, 50 mM Bis-Tris (pH 6.5) and 150 mM NaCl. After a second Ni-NTA purification, FOXO1 was applied to a S75 gel-filtration column (GE) equilibrated at 50 mM Bis-Tris and 150 mM NaCl. ETS1 purification involved an additional purification step using a Heparin column to remove DNA. The protein was eluted using a salt gradient from 50 to 2 mM NaCl. For NMR titration and backbone assignment experiments of FOXO1, the same purification steps were performed but Minimal medium M9 was used to isotopically enrich the protein. For $^{15}$N-labeled and $^{15}$N, $^{13}$C-labeled protein expression, $^{15}$NH$_4$Cl or $^{15}$NH$_4$Cl and $^{13}$C-Glucose were used as sole nitrogen and carbon sources, respectively. The final NMR and ITC buffer was 50 mM Bis-Tris, 150 mM NaCl, pH 6.5. The concentration of both proteins was measured using Nanodrop2000 and their absorbance at 280 nm. The purity of both proteins was determined by SDS gel electrophoresis and the ratio of absorbance at 260 and 280 nm.

**DNA oligonucleotide annealing**. Both oligo strands were added in equimolar ratios in a buffer containing 10 mM Tris pH 7.5, 50 mM NaCl and 1 mM EDTA. Afterwards, the samples were incubated for 5 min at 95 °C and cooled down 0.1°/s. The concentration of the aligned DNA molecules was determined by measuring absorbance at 260 nm using a Nanodrop2000.

**Isothermal titration calorimetry**. Titrations were carried out using either a MicroCal PEAQ-ITC or a MicroCal iTC200 calorimeter at 25 °C. All protein and DNA samples were dialyzed overnight at 4 °C against the same buffer containing 50 mM Bis–Tris, pH 6.5, and 150 mM NaCl. Concentrations used in each experiment are listed in Supplementary Data 3. For the FOXO1 titrations against the DNAs with or without ETS1 20 injections of titrant were made at 120 s intervals, while stirring at 750 rpm. The first injection was 0.4 μl, and 2 μl for the remaining nineteen. For the ETS1 titrations against the different DNAs 13 injections of 3 μl were done. While ETS1 was kept in the cell and DNA was added from the syringe, FOXO1 was always titrated from the syringe into the cell filled with DNA or DNA and ETS1 in equimolar ratios. Data were reduced with heat spikes from control and baseline corrected. The raw data integration, normalization and titration curve fitting was done using the MicroCal PEAQ-ITC analysis software provided by Malvern.

**NMR experiments**. All NMR measurements were performed at 298 K on an Avance III Bruker NMR spectrometer with a magnetic field strength, corresponding to a proton Larmor frequency of 600 MHz, equipped with a cryogenic triple resonance gradient probe head. Backbone resonance assignment of FOXO1 was achieved to a completion of 85% (excluding prolines) using $^{1}$H,$^{15}$N-HSQC, HNCA, CBCA(CO)NH and HNCACB triple resonance experiments[57] analyzed with CARA (http://cara.nmr.ch).

For all NMR titration experiments a series of $^{1}$H, $^{15}$N-HSQC spectra were recorded of $^{15}$N-labeled FOXO1 in absence or presence of equimolar unlabeled ETS1. Different DNAs (labeled as ω-none, and ω-high) were titrated always with the same series of molar equivalents (0.1, 0.2, 0.3, 0.4, 0.5, 0.6, 0.7, 0.8, 1.0, 1.2) to protein concentration (100 μM). As the DNA stock solution was highly concentrated (10 mM), the dilution effect was negligible but still taken into account. All spectra were processed using NMRPipe[58] and data analysis was performed using the program Sparky[59] for chemical shift perturbation analysis and CCPN for determining dissociation constants by fitting the fast exchange chemical shift perturbations vs. DNA concentration using $A(B + x - \sqrt{(((B + x)^2 - 4x)})$

as a fitting function[60]. Chemical shift perturbations were calculated with the following formula: $\mathrm{CSP} = \sqrt{\Delta H^2 + (0.2\Delta N)^2}$[61].

**Shape profiles calculation**. To quantify the contribution of DNA-shape in each TF binding position, we adapted an approach based on a conservative estimation of the performance change in $R^2$ after adding or removing a given shape feature in a given position[5]. Briefly, for each position $i$ in the TF binding model based on a consensus $k$-mer of length $k$, we calculated the minimum absolute change in the $R^2$ value ($\Delta R^2$) between two schemes: (1) increase after adding shape features in a sequence-only model ($1mer + \mathrm{shape}_i$) and (2) decrease after removing a shape feature in a shape-only model (shape $-$ shape$_i$)

$$a = \max\left(\Delta R^2_{(1mer, 1mer+\mathrm{shape}_i)}, 0\right) \quad (1)$$

$$b = \max\left(-\Delta R^2_{(\mathrm{shape}, \mathrm{shape}-\mathrm{shape}_i)}, 0\right) \quad (2)$$

From these two values, then the $\Delta R^2$ per position ($\Delta R^2_p$) is defined as

$$\Delta R^2_p = \min(a, b) \quad (3)$$

We considered tiled $k$-mers for the calculation of this value in CAP-SELEX data, so as the improvements are summarized by the median across all aligned positions in tiled $k$-mers. Similar to the trim-and-summarize $R^2$ comparisons in HT-SELEX data, we tested whether this scheme produces reliable agreements between $\Delta R^2_p$ profiles obtained between reference $k$-mers and their shorter tiled $k$-mers. For a number of trimmed positions equal to three, we have obtained a positive correlation distribution between $k$-mers and trim-and-summarize using shorter, tiled $k$-mers, which validates this approach for small trimming values (three or less) (Supplementary Fig. 4a, b).

**Clustering of shape profiles across SELEX datasets**. Comparing the similarity of $\Delta R^2_p$ values between all SELEX datasets requires alignment and assessment of similarity between binding models generated by $k$-mer tables of different length. To align such cases, we introduced an unbiased clustering scheme. Briefly, we applied a cubic spline interpolation to all shape profiles of a TF binding model to normalize them to 1000 points (function interp1d, from scikit-learn). Sometimes shape profiles can be mirrored and maximum $\Delta R^2_p$ values can be recovered in opposite positions across binding models (e.g., a TF binding model of length 9 with maximum $\Delta R^2_p$ value at position 3 contains its complementary model with maximum $\Delta R^2_p$ at position 7). To account for these cases, we inverted the shape profile if the improvement in maximum performance was located at positions after the respective profile mean (position 500).

Using these shape profiles with common length, we clustered them using a partitioning around medoids (PAM) routine implemented by the pam function (package cluster in R) with a defined number of clusters between 2 and 10. For each cluster, we calculated a cluster-specific TF family and TF enrichment as the odds ratio between the number of datasets for a TF family or a TF associated to this cluster versus the number of datasets for that same TF family or TF in all others clusters using Fisher's exact test. Significance $p$-values were corrected using the Benjamini Hochberg procedure. To select the reported number of clusters (five), we iteratively assessed the total number of TF and TFs families reported as enriched, stopping at the minimum clustering value that maximizes the number of raw $p$-values lower than 0.05 (see Supplementary Fig. S5a; see Supplementary Data 4).

**Analysis of Forkhead-Ets members using shape profiles**. To compare shape profiles of double and matched single TF datasets that have a common Forkhead TF member, we studied the $\Delta R^2_p$ values for FOXO1 and FOXI1, as the corresponding CAP-SELEX datasets are enriched in cluster 1 and most of their TF-pair combinations have an equivalent topology. To align TF binding models generated from CAP-SELEX and HT-SELEX, we used the consensus sequence motif of the Forkhead TF (listed in the reference $k$-mer) as an anchor point. Then, we maximized the number of matches between the Forkhead motif region and the reference consensus sequence across all composite motifs (FOXO1: RWMAAAC; FOXI1: RTMAAC). For ETS1, we used the GGAA pattern for alignment. HT-SELEX data for comparison was retrieved for FOXO1, FOXI1 and ETS1 using the available IDs in each case (Supplementary Data 4). Since these datasets capture short motifs, shape profiles can be generated using a single $k$-mer representing the consensus binding motif. Reference $k$-mers were used as in the CIS-BP database[11]. For aligning and comparing the profiles with the respective profiles for FOXO1, FOXI1 and ETS members we matched HT-SELEX $k$-mers to the respective composite $k$-mers reported for FOXO1 and ELK3 (ETS1 paralog), using the individual core motif for alignment, respectively.

FOXO1:ETS1 crystal structure is visualized in PyMOL[62] from PDB ID:4LG0[24], and enhanced with the PDIviz software[63]. Conservation of positive charge in the ETS1 residue 409 is calculated from the Pfam ID PF00178 (Ets-domain). DNA structure analysis is done using Curves +[64]. Comparison of shape feature differences between ω-none relative to ω/ω-high is done by Z-score normalization

of the observed Δshape (adapted from work by Wang et al.[65]) differences between positions 1–10, versus the expected ones in 1000 permutations of those positions.

**TF–TF motif enrichments in co-occupied ChIP-seq peaks.** ChIP-seq peaks used to assess TF–TF motifs in vivo were retrieved from the ReMap2 database[26]. Matched TF pairs from CAP-SELEX data were associated to ChIP-seq data when peaks for both TFs whenever possible. For obtaining common summit regions, we intersected the respective peak ranges centered around the peak summit with fixed length of 200 bp using bedtools (function intersect). These co-occupied regions are defined as the foreground set of peaks for each TF–TF pair. We discarded TF pair datasets that had less than 50 co-occupied peaks, recovering a total of 105 datasets. The background set, was defined as follows: For each foreground set, an equal number of 200 bp-long sequences with similar %GC content distribution was obtained from mappable hg19 regions (wgEncodeCrgMapabilityAlign36mer, downloaded from http://hgdownload.cse.ucsc.edu/goldenpath/hg19/encodeDCC/wgEncodeMapability), using the *BiasAway* software package[66].

To map motifs in these sequences, we prepared position weight matrices (*PWMs*) from the position frequency matrices provided in the CAP-SELEX dataset[10]. In both foreground and background sequences, we scored the best PWM motif hit per sequence, as the sequence that generates the highest score. These scores are used to define single feature models, labeled as PWM. Additionally, to define PWM + shape models[6], we extracted the DNA-shape features obtained for genomic regions aligned to all positions where a motif hit was obtained, using bwtool.

The ability of these features to separate foreground from background regions in each dataset was assessed as a classification task using Gradient boosting tree classifiers (XGBClassifier library[67]). Predictive features were independently centered and scaled. In a 10-fold cross validation scheme, the overall classification performance for each model and dataset was summarized as the median area under the receiver operating characteristic curve (ROC-AUC).

To assess the improvement of TF families in in vivo datasets when using *1mer* + shape vs. *1mer* models, we used their JASPAR family assignments, equivalently to the in vitro data analyses. For each TF family we specifically compared whether the ROC-AUC value differences between PWM + shape and PWM models (ΔROC-AUC) were significantly higher relative to all other datasets. Significance of the comparisons was assessed by a Wilcoxon rank sum test, and *p*-values were corrected using the Benjamini Hochberg procedure.

**In vitro and in vivo positional improvement correlations.** Similar to the $\Delta R^2_p$ calculation in SELEX data (see section: Shape profiles calculation), we generated ΔROC-AUC values per position for in vivo data (labeled as ΔAUC below), calculating changes in classification performance after addition (PWM + shape$_i$) and removal (shape - shape$_i$) of shape features in each position $i$ on in vivo models.

$$a = \max\left(\Delta\text{AUC}^2_{(\text{PWM,PWM+shape}_i)}, 0\right) \quad (1)$$

$$b = \max\left(-\Delta\text{AUC}^2_{(\text{shape}_i,\text{shape}-\text{shape}_i)}, 0\right) \quad (2)$$

$$\Delta\text{AUC}^2_p = \min(a, b) \quad (3)$$

In each matched TF–TF dataset with CAP-SELEX and ChIP-seq data, we aligned and compared $\Delta R^2_p$ values obtained from in vitro *1mer* + shape models and ΔROC-AUC per position values obtained from in vivo PWM + shape models using Spearman correlation. To estimate a false discovery rate (FDR) threshold for these correlation values, we scrambled correlation values in each model once and recalculated correlations.

**Forkhead-Ets cooperative *k*-mers in ChIP-seq data.** Selected sequences from our structural validation were mapped into co-occupied peaks to assess their enrichment versus single TF occupied peaks across TF pairs from the Forkhead and Ets families. Briefly, we mapped consensus sequences representing ω-none and ω motifs (GTAAACAGGAA and AACAACAGGAA, respectively), against Forkhead and Ets ChIP-seq in pairs, allowing up to one mismatch in each reported site. This threshold is chosen as it increases the recovery of sequences similar to each pattern, with a minimum overlap between hits in both categories. To compare the number of hits between co-occupied and single TF occupied peaks in each TF pair combination, we calculated the odds ratio (OR) of the number of sequences that do or do not containing either of these patterns in co-occupied versus single TF occupied regions:

OR = (a/b) / (c/d)

$a$ is the number of co-occupied peaks with the motif, $b$ is the number of co-occupied peaks without the motif, $c$ is the number of single TF occupied peaks with the motif, and $d$ is the number of single TF occupied peaks without the motif. We used a Fisher's exact test to assess the significance of these effect sizes across all assessed TF pairs, correcting *p*-values for multiple testing with the Benjamini Hochberg procedure.

**Modeling of associations between TF-pairs and ontologies.** Similar to the previous section, we prepared co-occupied and single TF occupied ChIP-seq regions for all TF pairs in the ReMap2 database, with full or partial match to

CAP-SELEX data. Full match indicates that both TFs in the ChIP-seq data are the same as in the CAP-SELEX data. Partial matches are two TFs that belong to the same TF family, and are annotated based on the idea that TF pair share composite motifs that are conserved within paralogs of the same family[46]. This knowledge can be used to extend the search to TFs of the same family for which no CAP-SELEX data is available. An example of this is provided with the TF pair between FOXO1 and ETV4: Both TFs are present in a CAP-SELEX dataset, and there are ReMap2 ChIP-seq peaks available for FOXO1, ETV1, ETV4, and ETV6. Thus, the TF–TF *k*-mers for FOXO1:ETV4 are used to scan co-occupied ChIP-seq regions of FOXO1:ETV4 (full match), FOXO1:ETV1 (partial match) and FOXO1:ETV6 (partial match). The full list of ChIP-seq pairs for TFs tested is available in Supplementary Data 7.

To assign co-occupied and single TF occupied ChIP-seq peaks to biological processes, we first used the software GREAT[27] with default parameters to map peaks to genes: Peaks are selected if located upstream of a Transcription Start Site (TSS) up to 5000 bp, downstream of a TSS up to 1000 bp, or nearby genes up to 1000 Kbp away from a TSS and in absence of other nearby genes. We then assigned genes to ontologies if they are listed in any of the three following ontologies: Gene Ontology Consortium (GO)[29,30], Human Phenotypes Ontologies (HPO)[31], and DISEASES database[32]. We only considered terms with at least 10 and no more than 1000 genes, to focus our analysis on terms with an amount of associated genes that facilitates interpretation.

Using this information, we sought to predict the membership of one or two TFs in a given ontology term, and use this as a proxy for their joint binding being associated to a biological function. This prediction is calculated using co-occupied TF peaks and counting the number of peak-gene pairs that are part of the ontology term in question. Co-occupied peaks are further stratified as cooperative (using the cooperativity *k*-mers) and non-cooperative. We assumed that a TF pair ($A$, $B$) is more likely involved in an ontology term (ont) based on the number of genes ($N_{\text{gene}}$) and peaks ($N_{\text{peak}}$) reported as part of that ontology term when using co-occupied ($A \cap B$) peaks and their peak-gene pairs. For any *ont* and ($A$, $B$) combination, $N_{\text{gene}}$ is lower or equal than $N_{\text{peak}}$, as multiple peaks can be mapped to the same gene ($N_{\text{gene}} \leq N_{\text{peak}}$).

The probability of a TF pair ($A$, $B$) to be associated with any *ont* $P(\text{ont} = 1|A,B)$, is directly proportional to $N_{\text{gene}}$ and $N_{\text{peak}}$.

$$P(\text{ont} = 1|A,B) \propto N_{\text{gene}}(\text{ont}|A \cap B)$$

$$P(\text{ont} = 1|A,B) \propto N_{\text{peak}}(\text{ont}|A \cap B)$$

To normalize $N_{\text{peak}}$ and $N_{\text{gene}}$ across all ontology terms and ($A$, $B$) combinations tested, we randomly sampled 200 times a number of unique regions equal to the observed number of $A \cap B$ from the original union of regions belonging to $A$ and $B$ ($A \cup B$)λ and recalculated decoy $N_{\text{peak}}$ and $N_{\text{gene}}$ values. From those we obtained mean ($\mu$) and standard deviation ($\sigma$) estimates for the expected $N_{\text{gene}}$ and $N_{\text{peak}}$ associated to that ontology in case of a false association. This is used to convert $N_{\text{peak}}$ and $N_{\text{gene}}$ into z-scores

$$Z_{\text{gene}} = \left(N_{\text{gene}}(\text{ont}|A \cap B) - \mu_{\text{gene}}\right)/\sigma_{\text{gene}}$$

$$Z_{\text{gene}} = \left(N_{\text{peak}}(\text{ont}|A \cap B) - \mu_{\text{peak}}\right)/\sigma_{\text{peak}}$$

Equivalently, when using TF–TF *k*-mers the association between an ontology (ont) and ($A$, $B$) is proportional to the number of peaks and genes obtained when using $A \cap B$ peaks, with a selection for the presence of TF–TF *k*-mers in those peaks. This is indicated as $N_{\text{gene},k}$ and $N_{\text{peak},k}$, where $k$ refers to the specific *k*-mer used

$$P(\text{ont} = 1|A,B,k) \propto N_{\text{gene},k}(\text{ont}|A \cap B, k)$$

$$P(\text{ont} = 1|A,B,k) \propto N_{\text{peak},k}(\text{ont}|A \cap B, k)$$

Similar to the previous Z-scores, we also normalize $N_{\text{peak},k}$ and $N_{\text{gene},k}$ into Z-scores using 200 random samplings

$$Z_{\text{gene},k} = \left(N_{\text{gene},k}(\text{ont}|A \cap B, k) - \mu_{\text{gene},k}\right)/\sigma_{\text{gene},k}$$

$$Z_{\text{peak},k} = \left(N_{\text{peak},k}(\text{ont}|A \cap B, k) - \mu_{\text{peak},k}\right)/\sigma_{\text{peak},k}$$

*K*-mer mismatch thresholds for each $A \cap B$, ont, and $k$ combination were defined so that they maximize $Z_{\text{gene},k}$ and $Z_{\text{peak},k}$ values. To do this, we allowed up to three mismatches in each *k*-mer to be mapped into a co-occupied peak, and recalculated the observed $Z_{\text{gene},k}$ or $Z_{\text{peak},k}$ values. When multiple *k*-mers for a pair ($A$, $B$) are available, we selected the one that gives the highest $Z_{\text{gene},k}$ and $Z_{\text{peak},k}$ values.

Integrating the resulting four Z-scores together, we defined an ontology association probability (OAP) as the probability of an ontology term associated to a TF-pair ($A$, $B$)

$$\text{OAP} = P\left(\text{ont} = 1|Z_{\text{peak}}, Z_{\text{gene}}, Z_{\text{gene},k}, Z_{\text{peak},k}\right)$$

This probability is modeled based on the four $Z$-scores obtained above using a logistic regression

$$OAP = \frac{1}{1 + e^{-\left(\beta_0 + \beta_1 Z_{peak} + \beta_2 Z_{gene} + \beta_3 Z_{gene,k} + \beta_4 Z_{peak,k}\right)}}$$

where $\beta_0$ defines the intercept and $\beta_1$ to $\beta_4$ the logistic regression coefficients for each $Z$-score. This model is limited by the availability of ChIP-seq data, and can be potentially extended as additional information is included. For example, if no ChIP-seq data is available then $k$-mers flanking Trascription Start Site can be considered, and $Z$-score calculations can be obtained using downsampling from all genes (Supplementary Fig. 6c).

**Benchmarking TF-pair and ontology associations.** To benchmark this modeling scheme, we tested its ability to distinguish ontology terms deemed as positive if one or both TFs in ($A$, $B$) are listed as genes of the term: TF1 or TF2 are TF-ontology relationships where one of the TFs is a gene member of that ontology term, whereas TF2 and TF2 contain both TFs as members of that ontology term. Note these two examples: (i) the HPO term HP:0002488 (Acute leukemia) includes the TF ETV6, but not the TF FOXO1, thereby the TF pair FOXO1:ETV6 has a TF1 or TF2 relationship to that particular ontology term (ii) The term HP:0002088 (Abnormal lung morphology) lists both MITF1 and FLI1 as gene members, and thereby the pair MITF:FLI1 has a TF1 and TF2 relationship to that term. Background terms are all ontology terms of which neither $A$ nor $B$ are members listed in those.

We assessed the predictive performance of the logistic regression using a full model with all $Z$-scores together ($peaks + kmers = Z_{peak}, Z_{gene}, Z_{gene,k}, Z_{peak,k}$) and variants with only the peaks or $k$-mer $Z$-scores ($peaks = $ only $Z_{peak}$ and $Z_{gene}$; $kmers = $ only $Z_{peak,k}$ and $Z_{gene,k}$). Performance metrics were defined by classification of *TF1 and TF2* terms versus *background* terms, or *TF1 or TF2* vs. *background* terms, independently (values in Supplementary Data 7). Positive to negative ratios for between *TF1 and TF2* and *TF1 or TF2* vs. *background* and total entries benchmarked in each ontology database are: HPO = 0.001 ($N = 99,828$) and 0.05 ($N = 1,715,112$); DISEASES = 0.001 ($N = 102,420$) and 0.04 ($N = 2,808,801$), and GO = 0.004 ($N = 166,104$) and 0.06 ($N = 906524$). Using a 10-fold cross validation approach, we trained models on nine portions of data and assessed the testing performance in the held-out portion, reporting the median ROC-AUC and area under the precision recall curve (PR-AUC) values using the trapezoidal rule. We compared significant improvement using an independent $t$-test between the ten testing performance metrics obtained in each model, correcting $p$-values using the Benjamini Hochberg procedure.

To define strong and weak TF pairs and ontologies, we shuffled the ontology labels ten times to assess the OAP mean score that falsely labeled positive (decoy) terms when fitting a model with those. We observed that the majority of mean decoys OAP values are no bigger than 0.1, with slight variations across ontologies and models. Assuming that OAP values 0.1 units higher than this empirical mean threshold are unlikely false associations, we used this threshold to separate signal from noise: Any TF pair ($A$, $B$) and ontology association labeled as a TF1 or TF2 or TF1 and TF2 with an OAP value 0.1 units greater than the mean of its decoys cases is considered a strong association. If multiple ontology terms for a TF pair satisfy these criteria, we visualize and discuss only the association with the highest OAP score (Supplementary Fig. 11b). For generating the Forkhead-Ets network, we additionally restrict all four $Z$-scores to be greater than zero.

**Time-to-treatment calculations.** We compared time-to-treatment metadata and RNA expression levels from Chronic Lymphocytic Leukemia patients ($N = 122$) from a Blood Cancer cohort[36]. Groups were separated using high and low expression levels for any TF pair of interest using the normalized counts median of given TFs. We compared between basic models, where both TFs have low expression (low/low) vs. models in which both genes have high levels (high/high), or models of the configuration high/low or low/high. Hazard ratios and confidence intervals are calculated using the survival package in R[68]. To correct for the immunoglobulin heavy chain variable gene (IGHV) and p53 mutation statuses we assessed an additional model that indicates if the patient has either of those factors reported as positive, as a single category ($N = 88$) (Supplementary Fig 6d). Models with patients with just one of the four combinations of these statuses are not reliable due to low sample numbers.

**Reporting summary.** Further information on research design is available in the Nature Research Reporting Summary linked to this article.

## Data availability

The backbone chemical shift assignment for FOXO1 (Forkhead domain) has been deposited at the Biological Magnetic Resonance Bank (BMRB) under the accession code 27894. All other relevant data supporting the key findings of this study are available within the article and its Supplementary Information files or from the corresponding author upon reasonable request. A reporting summary for this Article is available as a Supplementary Information file.

## Code availability

Scripts and instructions for reproducibility of the presented analyses are available at https://git.embl.de/grp-zaugg/coop-tf-binding. Input and output for relevant steps are available at http://www.zaugg.embl.de/data-and-tools/coop-tf-binding/.

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

## Acknowledgements

We are grateful to Kathryn Perez at the EMBL Protein Expression and Purification Facility for assisting with the ITC experiments. We acknowledge the feedback in parts of this work received from attendees of the meeting Rules of Protein-DNA Recognition held in Oaxaca, Mexico 2018, particularly from Remo Rohs. We also thank Natalie Romanov, Cecilia Perez-Borrajero, and Wolfgang Huber who provided feedback and discussion, and to Timothy Fuqua for constructive criticism of the manuscript. We also like to thank Anthony Mathelier and Benoit Ballester for early access to the ReMap 2018 database. Finally, we thank the funding provided by EMBL.

## Author contributions

I.L.I. and J.B.Z. conceived the study. I.L.I. designed the research and performed all computational analyses. N.M.H., S.A., and J.H. designed and conducted the experiments. B.V. and B.K. provided critical input on statistical analyses. I.L.I. and J.B.Z. interpreted the results and wrote the manuscript, with the critical feedback of all co-authors. J.B.Z. supervised the project.

## Competing interests

The authors declare no competing interests.
