## [Peer Review File · Nature Communications]

Reviewers' comments:

Reviewer #1 (Remarks to the Author):

This manuscript presents new computational analysis of the contribution of DNA shape recognition to cooperative transcription factor binding. As both TF-TF interactions and DNA shape are important areas of study in the field of gene regulation and DNA recognition, this paper will be of general interest to the field. The authors re-analyzed published CAP-SELEX data from the Jolma et al. 2015 paper, to identify DNA features that contribute to cooperativity.

The authors found that k-mer models that include either DNA shape features or 2- or 3- mers outperformed 1mer-only models at explaining DNA binding, and conclude from this that higher-order features are important for predicting TF cooperative binding. Adding higher order features to k-mer based predictions of DNA binding has been shown many times to improve model performance, so this finding makes sense. However, their claim that DNA shape contributes to cooperativity isn't a logical conclusion from these modeling results alone – higher order features also improve model performance for single TF families as well. So, this results section needs to be written more carefully. If a model for differentiating single vs cooperatively bound sites were improved by incorporating shape, then perhaps one could make the claim that DNA shape contributes to cooperativity, but from the way the model is described here, it does not seem to do that.

Furthermore, the authors need to show a clearer comparison of 1mer vs each of the more complex models, as Fig 1b compares just the 1mer-only model vs 1mer+shape model, while a comparison to a 1mer+2mer model needs to be shown. This is a key issue that impacts the conclusions of this paper. Importantly, if the 1mer+2mer model performs about as well as the 1mer+2mer+3mer or 1mer+shape models, then the improved performance could be explained as simply due to position interdependence at the di-nucleotide level, without involving shape readout. The authors should test to see if there is a statistically significant difference in the performance of the 1mer+2mer model vs the 1mer+2mer+3mer and 1mer+shape models. Looking at Fig S1c, the -1mer+2mer and 1mer+shape bars do not appear to be significantly different (their error bars overlap)). Also, it is unclear why the 1mer+2mer+3mer model outperforms 1mer+shape, and suggests overfitting.

The follow-up on the Forkhead-Ets interaction is nicely done, finding that the most highly cooperative sites are those with lower intrinsic forkhead binding. In particular, the idea that forkhead factors that don't show intrinsic FHL binding could recognize FHL site in complex with a cofactor. In figure 2c, the alignment between the 3 different sites is interesting – is that consistent with the best bound sites in the SELEX data? In the crystal structures of bispecific forkhead factors, it was shown that the C in the 6th position of the w-none site aligns with the C in the 4th position of the w-high, meaning the forkhead protein would adopt a different position with respect to the ETS protein. Related to this, how is spacing between the two binding sites evaluated in the modeling?

The result showing that position 409 in the Ets factor affects FoxO DNA binding affinity is very nice. Could the authors comment more on the mechanism by which they think this residue affects cooperative binding? Does it promote a certain DNA shape conformation that is optimal for forkhead binding?

On p. 7, lines 164-167, the authors refer to forkhead TFs binding w-none sequences on their own or w or w-high with an Ets partner, but this is by definition, since w and w-high were defined as cooperative and highly cooperative sequences. The authors should instead describe what are the sequence features of the sequences bound only cooperatively – do they have lower matches to the forkhead binding site motif?

p. 8, lines 190-191: "To generalize our cooperativity prediction to other FOXO1-Ets pairs..." –

expand the text to describe which forkheads and which partner TFs.

p. 9, lines 206-207: "in the case of FOXO1:ETS1, differences between high and non-cooperative DNA sequences were locally restricted (Fig 2c)" – Fig 2c doesn't actually show this. You would need to show the sequence and shape results beyond the composite motif. Similarly, lines 213-25: this doesn't appear to go beyond the core motif to the surrounding flanking nucleotides. Also, lines 217-290: the authors need to be clear in stating that their results about shape effects being localized to specific regions along the protein-DNA interface, which in their case is the core recognition motif, does not necessarily generalize to all TFs, just the forkhead proteins they looked at; for example, Gordan et al., Cell Reports, 2013 observed a shape effect for bHLH TFs in the flanking sequences extending beyond their recognition motif.

Fig 6: how does the "ontology association probability" which is shown on a scale of 0.2 to 0.6 relate to p-values? And can you calculate FDR q-values for the k-mer mean Z-scores?

The authors also analyze ChIP-seq data for these co-binding factors, to identify GO terms that are associated with the cooperation between the two factors.

The section on expression levels of FOXO1 and ETV6 as they relate to lymphoid leukemia patients is intriguing but does not really fit with the rest of the manuscript. Are there disease-associated mutations that affect the shape of the composite binding sites?

Minor comments:

Figure 1, R2 values overall are not high, even though they are consistently better when incorporating higher order variables. Are those low R2 values typical for these algorithms with selex data?

Line 143, change 'bi-specific Forkhead motif' to alternate forkhead motif or FHL motif

On p. 6, line 127, Fig S2c-d is incorrectly referred to, and should instead be Fig S1c-d. Also, I recommend this be moved to main body Fig 1.

p. 12, bottom paragraph: how much apparent variability in degrees of cooperativity would you expect by chance alone? Permutation testing may be useful in assessing this.

p. 13, line 314: p-value is significance, but what is the effect size? Also, what is the 'truth' as a positive control in this analysis? Is higher p-val more accurate, or is it simply an effect of small #s?

p. 14, line 331-2: A description of the "model-dependent signal-to-noise threshold on the association probabilities" should be provided in the main text.

No legend is provided for Fig S2c.

Guturu et al., 2013 is cited but not listed in the references.

The way they're aligned in figure 2C, the forkhead protein would be in a different register with respect to the ETS protein for the w-none and w than with the w-high. Are there spacing differences between motifs/most important kmers?

The manuscript needs proofreading for proper English grammar.

Reviewer #2 (Remarks to the Author):

In this densely written and data-rich manuscript, Ibarra and co-workers present a combined bioinformatic and experimental analysis of transcription factor (TF) cooperativity. Broadly speaking, they further analyzed the previously published CAP-SELEX database on human TF pairs (Jolma et al., Nature 527:384) to better identify DNA shape as a driver for cooperativity, and attempted to validate several aspects of their conclusions in vitro using NMR spectroscopy and isothermal titration calorimetry for ETS-Forkhead pairs and in vivo using phenotypes linked to TF associations. The manuscript is based extensively on bioinformatic approaches, for which I have little expertise, and thus it is hard for me to judge how much of an advancement it represents over past publications. Nevertheless, it remains a challenge to understand how TFs work in a cellular context to regulate gene expression, and this paper nicely highlights the complex role of DNA sequence in mediating TF cooperativity. This should be of interest to the readers of Nature Communications and I recommend that it be accepted for publication.

With that stated, I do have expertise in the biophysical chemistry of DNA-protein interactions, and suggest that the authors consider the following hopefully helpful comments on their experimental result.

- An intriguing conclusion of this paper lines with the statement on line 158 that the FOXO1-binding strength determined the level of cooperativity with ETS1. It would help if the authors clarified this statement in light of thermodynamic linkage. In such a coupled system, the free energy contribution for the FOXO1-ETS1-DNA cooperativity should add to the free energies of the individual protein-DNA interactions. Thus, the presence of FOXO1 should also increase the affinity of ETS1 for DNA (which unfortunately was not measured), and ETS1 cooperativity should increase the affinity of FOXO1 for both omega-none and omega-high sites. One possible explanation is that it is not the affinity of FOXO1 or ETS1 for a given site per se that matters. Rather, by changing DNA sequences, DNA-mediated cooperativity between ETS1 and FOXO1 may also have been changed. Separating such "direct" and "indirect" effects could be very difficult and may require studying a more extensive series of DNAs than the three listed in Fig 2c (which are somewhat biased having been selected from CAP-SELEX and HT-SELEX data sets to show or not show cooperativity).
- Given the limited information in the Methods section and Supplemental Material 3, it is very difficult to judge how the ITC experiments were carried out. What exact DNA oligonucleotides were used, how were they prepared, and how were concentrations of the proteins and DNA determined? Was the Ets1 included with DNA in the syringe (likely the case) or with the FOXO1 in the cell? What was the concentration of ETS1 and was the DNA saturated? Can the authors demonstrate that the measured heats were strictly from the FOXO1 binding the DNA or the saturated ETS1-DNA complex (and without contributions from changes in the ETS1-DNA interactions)? Also, in Figure S2a, the raw ITC data seem to rather noisy with significant changes even for endpoints (presumably heats of dilution) which likely impacts the reliability of the fit data. Furthermore, the raw data in the left panels of Figure S2a do not match the fit results give in the insets (likely reversed top/bottom). The reason I point this out is that it is extremely difficult to reliably measure Kd values for even simple protein-DNA interactions (reported literature results on the same systems often differ by orders of magnitude), and even more of a challenge to dissect cooperative interactions in multi-component systems. Minimally, this would require also measuring Kd values of ETS1 for DNA and the saturated FOXO1-DNA complex. Finally, the DNA sequences in Fig 2c (presumably used for the ITC studies) are truncated, e.g. an ETS1 site actually spans 9-12 basepairs. To what extent might this influence the measured Kd values and apparent cooperativity parameters?
- It might also be difficult for most readers to follow the simplified and perhaps overly optimistic discussion of the NMR-monitored titrations (lines 179 onward). In contrast to what is stated, in Fig

2e, K192 of FOXO1 shows slow exchange behavior when titrated with both omega-none and omega, without or with DNA (G230 in Fig S2c might be an example of what the authors are trying to conclude). In the case of omega-high, K192 of FOXO1 shows fast exchange with omega-high and only a slight change in lineshape when ETS1 is present. It is not clear that this is a switch from fast to intermediate exchange and hence indicative of cooperative binding (especially since the consequences of exchange on NMR spectra are dependent upon chemical shift differences, kinetics, and relaxation, with the latter certainly differing due to the molecular weight difference between the FOXO1-DNA and FOXO1-DNA-ETS1 complexes). Also, in Fig. 2f, the isotherms seems rather similar for FOXO1 binding omega-high DNA without or with ETS1. Taken together, it is unclear that there is any solid experimental evidence for ETS1-FOXO1 cooperativity with omega-high DNA. Similar to the ITC methods, it is also difficult to judge how the NMR experiments were carried out and interpreted. For example, Supplemental Material 3 lists G161 as one NMR-monitored signal, yet data is not shown and it is not identified in Fig S2b. G230 is also tabulated to have measurable chemical shift changes, yet the peaks shown in the spectra of Fig S2c seem unchanged over the titration series. Furthermore, K192 is tabulated to have chemical shift changes of 0.127 ppm and 0.223 ppm upon addition of 120 uM omega-high DNA without and with ETS1 present, respectively, yet in the lower panels of Fig 2e, very similar changes of ca. 0.12 ppm in the 1H dimension are seen.

- Finally, it is hard to be convinced by the authors' conclusion that Arg409 is the "driver" of FOXO1/ETS1 cooperativity. As noted above in the comments on thermodynamic cooperativity, it seems possible that the results of Fig 3f could simply reflect a reduction in the affinity of ETS1 for DNA due to the mutation, which then manifests as an apparently smaller increase in the affinity of FOXO1 for omega-medium DNA. The actual experimental conditions (including the sequence of omega-medium) and ITC data for this section of the paper have not been presented. Also, this highlights the need for additional experimental results, such as measuring the K_d values of WT and mutant ETS1 for DNA in the absence and presence of FOXO1. Although obviously well beyond the scope of this paper, it would also be interesting to compare the crystal structures of the FOXO1-ETS1-DNA complex with containing WT and A409A ETS1. In particular, if the author's conclusion regarding Arg409 holds true, then then conformation of the DNA should differ.

We want to thank the reviewers for their constructive and insightful critique of our manuscript.

Summary of the revised version:

We have addressed most of the reviewer's comments and highlighted them in the revised manuscript with (e.g. "Reviewer 1 - comment #1", **red text**). In addition, we highlighted the changes to the manuscript that are independent of reviewer's comments and mostly due to proofreading (e.g. "replace X with Y (proofread)", **magenta text**)

In addition to minor analyses and clarifications, we have added these **six** major analyses/experiments as requested by the reviewers (all described in more detail in the point-by-point response):

- more extensive comparison of predictive models using DNA shape vs. other high-order feature (**Fig 1c-d**)
- comparison of spacings between the DNA-binding domains for Forkhead and Ets when co-bound to the three tested DNA sequences (**Fig S2**), and analyses of DNA-structure features of those (**Fig S7**)
- additional analyses of the relative affinities of *k*-mers related to the three tested sequences in CAP-SELEX dataset with Forkhead+Ets TFs (**Fig S3b**).
- additional experimental evidence that the mutation in ETS1 does not alter its binding affinity and only affects cooperative binding of FOXO1 (**Fig S4**)
- comprehensive reporting of the ITC results (**Fig S4, S8**) results and Methods.
- comprehensive reporting of the NMR results (**Fig 2e, S5-6**) and Methods.

Detailed, point by point response:

Reviewer #1 (Remarks to the Author):

1. This manuscript presents new computational analysis of the contribution of DNA shape recognition to cooperative transcription factor binding. As both TF-TF interactions and DNA shape are important areas of study in the field of gene regulation and DNA recognition, this paper will be of general interest to the field. The authors re-analyzed published CAP-SELEX data from the Jolma et al. 2015 paper, to identify DNA features that contribute to cooperativity.

R: We thank the reviewer for acknowledging that this paper will be of general interest.

2. The authors found that k-mer models that include either DNA shape features or 2- or 3- mers outperformed 1mer-only models at explaining DNA binding, and conclude from this that higher-order features are important for predicting TF cooperative binding. Adding higher order features to k-mer based predictions of DNA binding has been shown many times to improve model performance, so this finding makes sense. However, their claim that DNA shape contributes to cooperativity isn't a logical conclusion from these modeling results alone – higher order features also improve model performance for single TF families as well. So, this results section needs to be written more carefully. If a model for differentiating single vs cooperatively bound sites were improved by incorporating shape, then perhaps one could make the claim that DNA shape contributes to cooperativity, but from the way the model is described here, it does not seem to do that.

R: We agree with the reviewer that we have to state our conclusion in the first section more carefully. In this revised manuscript version, we have toned down the statements regarding DNA-shape improvements, and primarily relate to high-order features. The reason we stick to shape in the end is because the improvements observed for Forkhead:Ets versus other datasets is only observed when including DNA shape features - and not higher order nucleotides ($1mer+2mer+3mer$, **Fig S1d** and **Reviewer comment #3**). This is further supported by our experimental validations.

Regarding the connection between TF-cooperativity and DNA-shape, we agree that we cannot claim with certainty that shape features explain cooperative binding specifically. In the revised version, we therefore adopt the term “co-binding” to be more precise in this results section. The term “cooperativity”, on the other hand, is adopted later on when considering both HT-SELEX and CAP-SELEX datasets (single and TF pairs, respectively). We believe that these changes in the text allow for a more accurate interpretation of our results.

“If a model for differentiating single vs cooperatively bound sites were improved by incorporating shape, then perhaps one could make the claim that DNA shape contributes to cooperativity, but from the way the model is described here, it does not seem to do that.”

We agree that a model comparing single vs. cooperatively bound sites would be a reasonable way to test the contribution of shape features in context of TF-cooperativity. However, this needs to be systematically explored and carefully modelled in a way that is beyond the scope of this paper.

Updates in Text/Figures:

We have:

- updated the title of the first results section (**line 82, underlined**): *“Quantitative modeling reveals contribution of higher order sequence features to TF co-binding”*.
- updated the goal of this section (**line 88, underlined**): *“To gain more insight into the mechanisms and uncover general rules that predict binding among the identified TF pairs, we hypothesized that features encoded in the DNA may contribute to the observed interactions. To test this, we devised a framework to predict the relative affinity of TF pairs based on DNA features using CAP-SELEX data. By ranking the importance of each DNA feature we could then identify those that potentially drive TF co-binding”*
- updated the main result statements (**line 124, underlined**): *“These results indicate that higher-order features are important for predicting TF co-binding”*.
- updated the conclusion at the end of the first section statement to reduce the emphasis on DNA-shape. (**lines 127, underlined**) *“Overall, regardless of whether high-order sequence features are interpreted as “DNA shape” or as “dinucleotide dependencies”, our results suggest their significance in the prediction of TF co-binding.”*
- updated the title of the second results section (**line 130, underlined**): *“Forkhead-Ets co-binding is linked to DNA shape features”*.
- Updated the first sentence of the second results section (**line 132, underlined**): *“We next sought to assess whether DNA shape/high-order features was important in driving co-binding between particular TF families.”*
- updated the following statement in Discussion to mention either base or shape readout, and not only DNA-shape, to relevant in our results (**line 410**): *“Thereby our models provide a platform for generating hypotheses regarding the potential consequences of disruptions in either base or shape readout in cooperative TF binding”*
- moved **Fig S1d** to the main results to illustrate the global observed performance for different model combinations.
- extended **Fig S1d** to include all the relevant scatter plots that compare feature combinations versus 1mer and in specific combinations: 1mer+shape+2merE2 vs 1mer+shape+2merE2, 1mer+shape+3mer vs 1mer, 1mer+shape+3merE2 vs 1mer+shape+2merE3, and 1mer+shape+3merE2 vs 1mer+2mer+3mer,

3. Furthermore, the authors need to show a clearer comparison of 1mer vs each of the more complex models, as Fig 1b compares just the 1mer-only model vs 1mer+shape model, while a comparison to a 1mer+2mer model needs to be shown. This is a key issue that impacts the conclusions of this paper. Importantly, if the 1mer+2mer model performs about as well as the 1mer+2mer+3mer or 1mer+shape models, then the improved performance could be explained as simply due to position interdependence at the di-nucleotide level, without involving shape readout. The authors should test to see if there is a statistically significant difference in the performance of the 1mer+2mer model vs the 1mer+2mer+3mer and 1mer+shape models. Looking at Fig S1c, the -1mer+2mer and 1mer+shape bars do not appear to be significantly different (their error bars overlap). Also, it is unclear why the 1mer+2mer+3mer model outperforms 1mer+shape, and suggests overfitting.

We have split the answer to this comment into 2 parts:

3.1 “Furthermore, the authors need to show a clearer comparison of 1mer vs each of the more complex models, as Fig 1b compares just the 1mer-only model vs 1mer+shape model, while a comparison to a 1mer+2mer model needs to be shown. “

R: We agree with the reviewer that it is important to compare extensively with other models such as 1mer+2mer models, or more complex variants, to rule out whether other feature types can explain our results, which we now provide in the revised manuscript version. We have updated **Fig S1c** and moved it to the main figures (**Fig 1d**, also requested by Reviewer 1 in a minor comment #1). Additionally, we provide pairwise comparisons and analyses of TF-families for all relevant models in **Fig S1d-e**.

3.2. “Importantly, if the 1mer+2mer model performs about as well as the 1mer+2mer+3mer or 1mer+shape models, then the improved performance could be explained as simply due to position interdependence at the di-nucleotide level, without involving shape readout. The authors should test to see if there is a statistically significant difference in the performance of the 1mer+2mer model vs the 1mer+2mer+3mer and 1mer+shape models. Looking at Fig S1c, the -1mer+2mer and 1mer+shape bars do not appear to be significantly different (their error bars overlap). Also, it is unclear why the 1mer+2mer+3mer model outperforms 1mer+shape, and suggests overfitting.”

We agree with the reviewer that 2mer features can encode dependencies at the nucleotide level. However, they can also describe stacking interactions between adjacent base pairs, which previous studies have highlighted to be indicative of DNA-shape features such as inter-base pair interactions [Zhou *et al.* 2015; Yang *et al.* 2017]. We acknowledge at this point that we can neither argue that shape alone explains the differences in performances relative to di-nucleotides, nor can we rule out the contribution of shape features completely.

One reason that explains the similarity in performance between 1mer+shape and 1mer+2mer models is that 1mer+shape models lack di-/tri-/shape features in the DNA flanks, unless added

explicitly as *2mer* or *3mer*. In this revised version, we therefore provide comparisons of *1mer+shape+2merE2*, *1mer+shape+2merE3* and *1mer+shape+3merE2* models, in which *2mer* and *3mer* features are included in the two (E2) or three (E3) positions flanking *k*-mers [Yang *et al.* 2017]. Both models show an improvement relative to the *1mer+2mer* model. Given that we took into account the flanking regions, and thereby provided a common baseline, we can conclude from this analysis that *shape* outperforms *2mer* features.

Regarding the comparison between *1mer+2mer+3mer* and *1mer+shape* models (or *+2merE2/3merE2*), we observe that the *1mer+2mer+3mer* model consistently performs better in CAP-SELEX data. We acknowledge that the studies from Yang *et al.* and Zhou *et al.* and other previous studies have compared both types of models (*1mer+2mer+3mer* vs. *1mer+shape* or variants of it) in SELEX and other experimental platforms and reported similar improvements for these models. Yet in CAP-SELEX data we observe that the *1mer+2mer+3mer* model significantly outperforms the *1mer+shape* model, with and without *3merE2* features (see Figure 1d). We have explored confounders that could explain this discrepancy; however, we cannot identify any data-processing step that accounts for this result (mean R^2 vs. median R^2 for reporting test R^2 , palindromic sequences and averaging of the feature vectors, count-based filtering of *k*-mers and datasets). We rule out overfitting in our models as raised by the reviewer, because we are reporting median R^2 metrics based on a 10-fold cross validation for all model setups (i.e. the median of ten R^2 values obtained in 1/10 held-out test sets, using 9/10 of data for model training). One reason that could partially explain the differences between the *1mer+2mer+3mer* model versus the *1mer+shape* model is that shape features are based on pentamers that cannot map to the two flanking positions, as pointed out in the previous comment. However, we tested that after addition of *3merE2* features we still get slightly lower performances relative to *1mer+2mer+shape* models. At the same time, it is noteworthy that we were able to successfully reproduce the results from Yang *et al.* on HT-SELEX datasets (Reviewer figure 1). This does not seem to hold true for CAP-SELEX datasets, in an equivalent implementation.

It is possible that the *3mer* model considers features beyond single TF-binding modes (e.g. alternative TF-TF binding modes, or even sequencing biases), which could explain the stronger performance versus models that include *shape* features only.

Nevertheless one line of evidence that demonstrates the importance of *shape* for cooperative binding is that the improvement of TF-TF binding predictions is similar for all families when using the high-order sequence models (*1mer+2mer+3mer*), whereas the *shape* containing models (*1mer+shape* or *1mer+shape+3merE2*) show a distinct improvement specific to the Forkhead-Ets family combination only. We have included all boxplots highlighting those improvements (see **Fig S1e** and **Supp Material 1**). Additionally, the crystal structure of the FOXO1-ETS1 complex reveals a specific interaction with a DNA minor groove that agrees well with the predicted positional improvements (“shape profiles” in this work) of the TF-binding models of Forkhead+Ets, and minor groove features in the same position for ω -none/ ω (see comment #6). Thus, our work indeed provides evidence for the case-specific importance of shape readout.

Reviewer figure 1. $1mer+shape+3merE2$ and $1mer+2mer+3mer$ performances in HT-SELEX and CAP-SELEX data. (left). Reanalysis of HT-SELEX shows reproducible R^2 -values in $1mer+shape+3merE2$ versus $1mer+2mer+3mer$ (adapted from [Yang et al. 2017]). (right) In our equivalent setup for CAP-SELEX data, we have observed a similar trend but with more additional datasets having better performances when comparing the same models as on the left side.

Updates in Figures/Text:

We have:

- added comparisons for relevant models in the main figure barplot (**Fig 1d**): $1mer+shape$, $1mer+shape+2merE2$, $1mer+shape+3merE2$, and $1mer+2mer+3mer$. In addition to that, asterisks indicating significant statistical differences between models with increasing performance are included in the main figure.
- added scatter plot for relevant pairwise comparisons between models (**Fig S1d**).
- added additional boxplots showing improvements by TF family between augmented and $1mer$ models (**Fig S1e**).
- updated this goal sentence (**line 132**): “We next sought to assess whether DNA shape/high-order features was important in driving cooperativity between particular TF”
- updated the last sentence of the first results section into the second section, to better highlight our rationale for using $1mer+shape$ models (**line 142, underlined**): “No such family-specific bias was observed when using any of the other high-order models. And despite their overall R^2 -values being higher than in the $1mer+shape$ model (Fig 1b; S1d), the median R^2 -values for Forkhead+Ets datasets were highest in the $1mer+shape$ models (Fig S1e). Additionally, this is interesting because crystallographic studies have demonstrated that DNA shape varies across Forkhead members [Li et al 2017]. In addition, Ets binding predictions have shown improvements by DNA-shape features”

[Yang et al. 2017]. For these reasons we decided to specifically investigate the improvements obtained by including DNA shape and therefore focus the analysis in the following sections on the 1mer+shape models.”

Beyond the answers for each part, we want to clarify that comparisons with previous results from other authors have proven difficult because the code for model setups and performance calculations is not available. We think that reproducing their results extensively is beyond the scope of this work. We have included all the processed data and code used to obtain the multiple linear regression R^2 values. This will increase the reproducibility of our results and allow the community to re-use our code for other datasets as well.

4. The follow-up on the Forkhead-Ets interaction is nicely done, finding that the most highly cooperative sites are those with lower intrinsic forkhead binding. In particular, the idea that forkhead factors that don't show intrinsic FHL binding could recognize FHL site in complex with a cofactor. In figure 2c, the alignment between the 3 different sites is interesting – is that consistent with the best bound sites in the SELEX data?

R: We thank the reviewer for this comment. Briefly, the alignment of the three sites is consistent with the best bound sites in the CAP-SELEX datasets. Here we describe an analysis that supports this (included as **Fig S3b** in the updated manuscript version):

To compare best bound k -mers in SELEX data we computed for each k -mer a relative enrichment in each selection round versus their expected value given an input library (round zero). For each 13-mer that matches, we can estimate its relative affinity using the observed relative affinities of 10-mers that contain patterns within the 13-mer, averaging those into a single metric. We have done this for all CAP-SELEX datasets, and compared the three presented 13-mers related to Forkhead+Ets sites (ω -none: GTAAACAGGAAGT/ ω : AACAAACAGGAAGT/ ω -high: ACGCACCGGAAGT), summarizing their estimated relative affinities in each dataset (**Reviewer Figure 2**).

Between non-Forkhead and Forkhead containing datasets, there are differences in the distribution of relative affinities for 10-mers related to ω -none and ω . However, in these cases the effect is not so apparent for ω -high (FHL site). When comparing SELEX datasets for Forkhead+Ets relative to only Forkhead (no Ets), or only Ets (no Forkhead), we observed an increase in the median relative affinity of all three composite sites. In the particular case of FOXO1:ELK1, which we highlighted in Fig S3b, the averaged relative affinities are 0.84, 0.74 and 0.37 for ω -none/ ω / ω -high, respectively (circle point in last boxplots group).

This analysis hints at FHL sites being among the best bound sites in CAP-SELEX data for Forkhead+Ets families. We have included this analysis as a supplementary figure (**S3b**). We feel that an extensive analysis of both multiple SELEX and *in vivo* data would distract from the main messages of the paper and thus be out of scope here. However, it would be interesting to follow up on this in a separate study.

Reviewer Figure 2. Relative affinities of k -mers related to examined Forkhead-DNA binding sites. Average relative affinities from CAP-SELEX data for 10-mers within 13-mers related to ω -none/ ω / ω -high. From left to right: other datasets (not including Forkhead, nor Ets), Ets without Forkhead, Forkhead without Ets and Forkhead+Ets datasets. Dots in the Forkhead+Ets group indicate averaged relative affinities for FOXO1:ELK1 using related 10-mers. Asterisks indicate Wilcoxon rank sum test between Forkhead+Ets group and other groups (two-sided), with Benjamini Hochberg correction.”

Updates in main text/figures:

We have:

- included **Reviewer Figure 2** and its Legend as a Supplementary Figure panel (**Fig S3b**).
- added the following sentence in the main text (**line 175**): “Despite a low affinity for ω -high in microarray data, we observed higher median relative affinities in CAP-SELEX datasets with both Forkhead and Ets factors than in other datasets (Fig S3b; Methods)”.
- updated the Methods section to indicate the following update (**line 872**): “For CAP-SELEX datasets, 13-mers connected to those three sequences were compared by averaging the relative affinities of all 10-mers contained within those into a single value.”

- replaced the title of that Methods section including this analysis (**line 867**): “PBM data analysis” with “Comparison of *k*-mers with cooperative Forkhead+Ets sites in PBM and CAP-SELEX”

5. In the crystal structures of bispecific forkhead factors, it was shown that the C in the 6th position of the ω -none site aligns with the C in the 4th position of the ω -high, meaning the forkhead protein would adopt a different position with respect to the ETS protein. Related to this, how is spacing between the two binding sites evaluated in the modeling?

R: We thank the reviewer for pointing this out. It is indeed an interesting observation that we have followed up on by comparing two structures for Forkhead DNA-binding domains bound to Fkd/Fhl sites, and one co-crystallized with ETS1. In ω -none and ω sites (Fkd) FOXO1 and ETS1 would have a spacing similar to the one observed in the crystal structure (**Reviewer Figure 3a**), but in the Fhl site the spacing would be different, as highlighted by the reviewer. This is explained by structural alignment of ω -none, ω and ω -high and the labeling of start and end positions that should be structurally equivalent to the crystal structure, Fkd:Rtaaac**A**; Fhl;**G**acg**C** (**Reviewer Figure 3b-c**). When the three sequences and the sites are aligned and the first positions are indicated (**R** and **G** for Fkd and Fhl, respectively), there is a (+1)-shift of the Fhl site towards the upstream region.

The spacing between TFs in our setup is based on reference *k*-mers which have been found enriched in the CAP-SELEX study by Jolma *et al.* 2015. Those are assumed to be good starting points to model sequences with a certain level of similarity. Given low level of mismatches between the sequences, this should recover binding patterns with no changes in TF spacing. The recovery of *k*-mers with alternate spacing such as ω -high is done by comparison of the relative affinities between matched HT-SELEX and CAP-SELEX data (**Fig 2b**). These can also be recovered by modeling relative affinities allowing more mismatches. We have not extensively studied the usage of multiple reference *k*-mers or parameters that consider multiple spacings jointly.

Updates in text/figures:

We:

- added the **Reviewer Figure 3** and its legend as a new figure (**Fig S2**).
- added a sketch of possible conformations in **Fig 2c**, next to the respective sequences.
- added a short statement and a reference to this new Figure in the main text (**line 179**): “Interestingly, the crystal structure suggests an alternate spacing between Forkhead and Ets DNA-binding domains for ω -high (Fig S2) [Rogers *et al.* 2019]”. The reference to Rogers *et al.* is included in the References List.

Reviewer Figure 3. Forkhead and Forkhead-Ets structures with Fkd and Fhl DNA sequences. (a) FOXO1:ETS1:Fkd (green/orange and gray) crystallized complex (PDB ID = 4bl0). DNA nucleotides in red indicate start and end positions of RYAAACA motif (Fkd), and yellow indicates intermediate nucleotides. (b) Structures of FOXO1:Fkd (green/black) and FOXN3:FHL (magenta/gray) complexes (PDB ID = 3co6 and 6cnm [Rogers *et al.* 2019], respectively). (c) Superposition of FOXO1:ETS1:Fkd complex with FOXO1:Fkd ((i) +(ii)) and FOXN3:Fkd ((i) +(iii)) using Forkhead DNA-binding domain as a reference for structural alignment (function align in PyMOL). For FOXN3:FHL, red nucleotides in DNA indicate start and end of GACGC site, and yellow indicate intermediate nucleotides. (c) Sequences ω -none, ω and ω -high as studied in this work. Nucleotides in red indicate start and ending positions for DNA binding domain binding regions (Fkd for ω -none and ω , FHL for ω -high). The start nucleotide for ω -high is shifted by one position with respect to ω / ω -none (expected no shift), and the Ets binding region is aligned, suggesting a (+1)-spacing between Forkhead-Ets DNA-binding domains in ω -high with respect to ω -none/ ω .

6. The result showing that position 409 in the Ets factor affects FoxO DNA binding affinity is very nice. Could the authors comment more on the mechanism by which they think this residue affects cooperative binding? Does it promote a certain DNA shape conformation that is optimal for forkhead binding?

R: This is indeed an interesting question. We tackled it by looking into the predicted DNA-structures of ω -none, ω , and ω -high sequences in each respective position. For control purposes we included scrambled versions of the DNA-sequences. We measured the magnitude of change in structural features relative to control sequences as a Z-score (Reviewer Figure 4, included as a new Fig S7). Some structural features tend to be different relative to the expected

values in the scrambled sequences, such as the propeller twist for example. However, we do not see consistently large changes overall.

When focusing only on the position where we observed the improvement peak (**Figure 3c**), we observed a local minimum of minor groove width in the crystal structure of FOXO1:ETS1:DNA for this particular position. This observation correlates with the decrease in minor groove width for the expected free DNA values (ω -none, ω , ω -high), respectively. This in turn might suggest a connection between cooperativity and small minor groove width in that position.

Reviewer Figure 4. DNA structural features for studied Forkhead-Ets binding sites. Minor groove width (MGW) values are highlighted for each of the studied Forkhead-Ets DNA binding sites. Values are shown as heatmap (top) and line plots (bottom). Black line shape parameters obtained for the DNA molecule within the FOXO1:ETS1:DNA complex (PDB ID = 4lg0), using Curves+ [Blanchet *et al.* 2011]. Highlighted region indicates positions where Forkhead+Ets performances changes in CAP-SELEX data were the highest.

At the moment we speculate that the mechanism entails an interaction between ETS1 and FOXO1 partially mediated by the Arginine residue. We cannot describe the individual steps in the formation of the complex, but our data support that at some point the Arginine residue binding increases the potential for FOXO1 binding, improving its overall affinity. As suggested by reviewer 2, the loss of cooperativity seen in ITC when mutating R409 to alanine, could also be due to weaker ETS1 binding. However, in our revisions, we measured ETS1-R409A alone and the affinity is in the same order of magnitude as for wildtype (see reviewer 2 comment 11).

Updates in text/figures:

We have:

- included an extended version of **Reviewer Figure 4 (Fig S7)**, with an according legend.

- included a reference to this figure in the main text (**line 271**): “*This agrees with the relevance of Minor Groove Width features for the binding predictions in our models (Fig S10c; Fig S7a)*”
- included a description of this analysis in Methods (**line 1001**). “*DNA structure analysis is done using Curves+ [Blanchet et al. 2011]. Comparison of shape feature differences between ω -none relative to ω/ω -high is done by Z-score normalization of the observed Δ shape (adapted from [Wang et al 2018]) differences between positions 1 to 10, versus the expected ones in 1000 permutations of those positions.*” A reference to Blanchet et al. was already included in the reference list.

7. On p. 7, lines 164-167, the authors refer to forkhead TFs binding ω -none sequences on their own or ω or ω -high with an Ets partner, but this is by definition, since ω and ω -high were defined as cooperative and highly cooperative sequences. The authors should instead describe what are the sequence features of the sequences bound only cooperatively – do they have lower matches to the forkhead binding site motif?

R: We thank the reviewer for this comment, and we have amended that phrasing in our manuscript. In this version we also describe the sequence features that can be used to compare ω -none/ ω/ω -high. As in Fig 2b, the predicted relative affinity of FOXO1 alone is negatively correlated with the increased cooperativity. In addition to that, there is a negative correlation between mismatches in the Forkhead binding site and the Forkhead-Ets observed cooperativity. We further observe a positive correlation between Ets matches in its binding site and the observed cooperativity. This was not explicitly shown, but now it is included in the main panels and stated in the main text, including the correlation coefficients between estimated cooperativity and the number of mismatches in the Forkhead and Ets regions.

Reviewer Figure 5. Association between motif matches in Forkhead:Ets composite sites and cooperativity. Correlations between number of motif matches and cooperativity for (*left*) Forkhead and (*right*) Ets binding sites in 15-mers. ρ and P indicate Spearman correlation and p-

value, respectively. Color indicates median cooperativity value).

Updates in main text/figures:

We have:

- added **Reviewer Figure 5** to the main figures (**Fig 2b**).
- referred to this figure in the main text of the revised version (**line 177, underlined**): “These results and the negative correlation between Forkhead motif matches and cooperativity (Fig 2b) suggested that Forkhead TFs can bind to ω -none sequences on their own by recognizing a strong Forkhead binding site while relying upon allosteric interactions with their Ets partner for recognizing ω (and possibly ω -high) sequences by forming a cooperatively bound ternary complex.”

8. p. 8, lines 190-191: *“To generalize our cooperativity prediction to other FOXO1-Ets pairs....” – expand the text to describe which forkheads and which partner TFs.*

R: We have included the names of the TFs present in those CAP-SELEX datasets in the main text.

Updates in text:

- (**line 219, underlined**) “...In CAP-SELEX, however, relative affinities for ω -none and ω were similar for the majority of datasets comprising FOXO1 paired with Ets (ELF1; ELK1/3; ETV4/5), Homeodomain (HOXB13) and GCM (GCM1) members...”

9. p. 9, lines 206-207: *“in the case of FOXO1:ETS1, differences between high and non-cooperative DNA sequences were locally restricted (Fig 2c)” – Fig 2c doesn’t actually show this. You would need to show the sequence and shape results beyond the composite motif. Similarly, lines 213-25: this doesn’t appear to go beyond the core motif to the surrounding flanking nucleotides. Also, lines 217-290: the authors need to be clear in stating that their results about shape effects being localized to specific regions along the protein-DNA interface, which in their case is the core recognition motif, does not necessarily generalize to all TFs, just the forkhead proteins they looked at; for example, Gordan et al., Cell Reports, 2013 observed a shape effect for bHLH TFs in the flanking sequences extending beyond their recognition motif.*

R: We agree with the reviewer that the precision in the description of our results needs to be increased, regarding the limitations and lack of analysis beyond the core motif. We formulate our results on Forkhead-Ets more precisely now, and do not generalize our findings to all TFs (see updates in Text).

(1) Regarding core features alone, in the original manuscript we made a comparison between the positions, where prediction performances change the most. In the case of Forkhead+Ets

datasets, we compared specific cases Forkhead members (FOXO1, FOXI1) with their partner TFs using matched HT-SELEX data for each of them alone. We observed that the single TFs alone cannot explain the observed improvement in the region where we found the strongest improvements by DNA shape for Forkhead+Ets. (**Fig 3c**; **Fig S10d-e**), and we consider these results to be evidential of specific positions within the core site in Forkhead+(Ets/Homeodomain/GCM) composite sites. Importantly, these local improvements are shown for all analyzed datasets in **Fig 3b**, and suggest that are just limited to Forkhead+Ets. However, we could not find any other TF significantly enriched for a cluster of shape profiles (**Fig S10b**), which is why we focused on these. Studying multiple cases to the detail we did is beyond the scope of this study.

(2) Unfortunately, CAP-SELEX data does not allow us to study whether features beyond the core motif are relevant in the predictions because the length of k -mers that consider core + flanks generate sparse counts in the CAP-SELEX datasets and make the relative affinity estimates unreliable for most TFs in these data. We can speculate that by the conformational changes in the Forkhead wings proximal to the flanks due to interaction with co-factors, the flanking regions might indeed play a role. We cannot, however, quantify such a scenario reliably in our setup (*tiled k-mers*).

Updates in text/figures:

We have updated the following lines of the Results section:

- **(line 230)**: *“in the case of FOXO1:ETS1, differences between high- and non-cooperative DNA sequences were related to changes in the Forkhead site”*
- **(line 244)**: *“To globally explore the positional effects of shape profiles in the core motifs”*

We have included the following statements in the Discussion of the manuscript:

- **(line 419)**: *“Although these results cannot be generalized to all TFs and are limited to Forkhead:Ets pairs, they highlight the relevance of our modeling for detecting such specific interactions in the core motif, which we also observed in the CAP-SELEX data for other pairs, yet with weaker enrichments”.*
- **(line 433)**: *“Despite data sparsity impeding the generation of models including features beyond the core motifs [Gordân et al 2013]”*

Finally, we have included Gordân et al. 2013 in the references list (**line 546**).

10. Fig 6: how does the “ontology association probability” which is shown on a scale of 0.2 to 0.6 relate to p-values? And can you calculate FDR q-values for the k-mer mean Z-scores?

R: In this version we have calculated FDR thresholds for our ontology association probabilities by scrambling and recalculating the obtained probabilities ten times using the original model, in each ontology and benchmark combination (TF1 and TF2 versus background and TF2 or TF2 versus background). The obtained thresholds are indicated in **Supp Material 7**. Using the HPO database as an example, the FDR thresholds are 50/20/10 for association probabilities of

~0.20/0.24/0.28 when classifying “TF1 and TF2” and ~0.44/0.50/0.53 when classifying “TF1 or TF2” terms versus background terms. We have included an FDR threshold of 10% for both types of positive associations, in the figure colorbar (**Fig 6a**).

Regarding the calculation of an FDR threshold for the k -mer mean Z-scores, this is something that we have previously attempted for all features independently. However, we must point out that FDRs using only k -mer based features are high (FDR > 50%), and therefore unreliable. The reason for this is that k -mer only models (only using count features related to k -mers, 3 and 4), have the lowest Precision-Recall AUCs, in comparison to *peaks* or *peaks+k-mers* models (Fig 5c and S6a). Our recommendation is to use ontology association probabilities as a first filter, taking into account both co-occupied peaks and composite k -mer information. Then, it is possible to interpret the mean k -mer Z-scores that are retrieved from those top-associations.

Updates in text/figures:

We have:

- included a label to indicate FDR thresholds in the colorbar legend of **Fig 6a**: “FDR=10%”, which indicate the minimum score necessary to report “TF1 and TF2” and “TF1 or TF2” associations.
- updated the Legend of **Fig 6a** to point this out “..FDR thresholds in color scale indicate “TF1 and TF2” confidence for association probabilities obtained using the HPO ontologies database (extended thresholds in Supp Material 7)...”.
- updated these threshold in the **Supp Material 7** (sheet “FDR”).
- included the following text in Discussion (**line 452, underlined**): “*In fact, the knowledge of cooperative k -mers jointly used with ChIP-seq data translates into better TF-ontology predictions and could thus increase the extent of our functional knowledge on cooperative TF binding and its underlying biology (Fig 7)...*”

11. The authors also analyze ChIP-seq data for these co-binding factors, to identify GO terms that are associated with the cooperation between the two factors. The section on expression levels of FOXO1 and ETV6 as they relate to lymphoid leukemia patients is intriguing but does not really fit with the rest of the manuscript.

R: We disagree with this comment and want to stress that our validation for FOXO1/ETV6 expression fits well with the rest of the manuscript. We argue that the association of FOXO1/ETV6 and overall survival in leukemia works as an external validation of one of our top-predictions for which data is available. We acknowledge that giving a lot of emphasis on it can be a distraction, but removing it would reduce the expected applicability we envision for this approach. Specifically, the provided prediction is supported by (1) our ontology association probabilities, (2) cancer expression data, and (3) recent reports indicating a gene fusion for FOXO1/ETV6 [Stengel *et al.*, 2018], (highlighted in discussion). The observation of this trend after controlling for P53 and IGHV mutation status further validates its relevance.

12. Are there disease-associated mutations that affect the shape of the composite binding sites?

R: We thank the reviewer for bringing this point to our attention. To identify possible associations of binding motifs with disease-associated genetic variants, we have mapped the binding motifs **RWAAAGAGGAA** (ω -none/ ω) and **GACGCNNNGGAA** (ω -high) to the GWAS catalog (NHGRI-EBI, build 2018). In the table below, up to ten SNP associations per motif are shown, with (1) less than 2 mismatches to the query motif, (2) a ChIP-seq peak for at least one Forkhead or Ets factor, or (3) QTL support in lymphoblastoid cell lines [Grubert *et al.* 2016].

rsID	seq.fwd	seq.rev	ref/alt	coordinate	strand	matches	phenotype	chip.seq			Δ shape [Wang et al 2018]			
								Ets	Fkd	QTL	MGW	HELT	PROT	ROLL
rs66782572	gAcgcccggggca	tgccccggcgTc	A/G	chr3:52567617	+	Fhl:8	Hemoglobin concentration	1	0	1	1.4	4.7	11.7	3.9
rs10496435	tttcaacgcGtc	gaCgcgttgaaa	G/A	chr2:110929173	-	Fhl:8	Neuroticism	0	1	0	3.1	4.7	9.2	12.2
rs7708584	ctccAttgcGtc	gacGcaaTggag	A/G	chr5:153543466	-	Fhl:8	Body mass index (adult)	0	1	0	0.8	2.0	1.2	3.2
rs79881201	ctccaaggCgtc	gacGccttgag	C/T	chr5:110427795	-	Fhl:8	Sum eosinophil basophil counts	1	1	0	0.5	2.6	1.4	2.2
rs1131510	gaCgctcatgac	gtcatgagcGtc	C/T	chr10:35299085	+	Fhl:7	Colorectal adenoma (advanced)	0	1	0	0.5	2.2	6.4	2.4
rs7164479	ggcccCgcgctc	gacgcGggggcc	C/T	chr15:79123054	-	Fhl:7	Coronary artery disease (myocardial infarction, percutaneous transluminal coronary angioplasty, coronary artery bypass grafting, angina or chronic ischemic heart disease)	1	0	0	0.9	3.0	0.6	2.7
rs55853698	cTctgctgcGtc	gacgcagcagAg	T/G	chr15:78857939	-	Fhl:7	Small cell lung carcinoma	1	1	0	1.3	1.9	6.7	9.1
rs4537545	gaCgccaagca	tgcttcggcGtc	C/T	chr1:154418879	+	Fhl:7	C-reactive protein	0	1	0	0.4	1.9	1.7	2.0
rs35936514	gacgCatcctaa	ttaggatGcgtc	C/T	chr10:126244970	+	Fhl:7	Major depressive disorder	0	1	0	1.7	2.3	5.0	11.5
rs12044149	tagcaacGcgtc	gacgCgttgcta	G/T	chr1:67600686	-	Fhl:7	Psoriatic arthritis	1	0	0	1.0	2.4	6.7	3.0
rs10467147	gtaaacagGaa	ttCctgtttac	G/A	chr12:40767362	+	Fkd:11	Obesity-related traits	0	1	0	0.5	2.6	1.3	2.4
rs1330225	ttcctgTttgc	gcaaAcaggaa	T/C	chr1:106835943	-	Fkd:10	Blood pressure measurement (low sodium intervention)	1	1	0	0.7	1.1	4.8	3.7
rs7230711	ttcctggttAt	aTaaccaggaa	A/G	chr18:48690619	-	Fkd:10	Immune response to anthrax vaccine	0	1	0	1.2	1.9	5.7	7.9
rs4764039	taCctgtttac	gtaaacagGta	C/T	chr12:14064461	-	Fkd:10	Total ventricular volume	0	1	0	0.6	4.3	6.8	2.6
rs2145623	agaaaacagGaa	ttCctgtttct	G/C	chr14:35839236	+	Fkd:10	Chronic inflammatory diseases (ankylosing spondylitis, Crohn's disease, psoriasis, primary sclerosing cholangitis, ulcerative colitis) (pleiotropy)	1	1	1	0.5	2.1	5.0	2.3
rs7582141	ataaacagGaaa	tttCtgtttat	G/T	chr2:159899489	+	Fkd:10	Response to radiotherapy in prostate cancer (toxicity)	0	1	0	0.7	4.9	11.8	2.4
rs1472750	ttcaTgtttac	gtaaacAtgaa	T/C	chr10:115235114	-	Fkd:10	Myopia (pathological)	0	1	0	0.6	3.9	8.5	2.4
rs115500520	attaAcaggaa	ttcctgTtaat	A/T	chr4:164447657	+	Fkd:10	Itch intensity from mosquito bite adjusted by bite size	0	1	0	0.5	4.3	9.7	4.0

Reviewer Table 1. Best associations to GWAS SNPs using Fkd:Ets (ω -none/ ω , $N=10$) and Fhl:Ets (ω -high, $N=8$) composite binding sites as seeds for motif matching.

To describe the extent by which shape features in the composite sites are affected by disease-associated variants, we also added the averaged “delta shape” value for each GWAS variant as defined by Wang *et al.*, 2018. This value quantifies the magnitude of change for Minor Groove Width (MGW), Helical Twist (HELT), Propeller Twist (PROT), and Roll (ROLL), between reference and alternative alleles. We think that the assessment of these disruptions being significant for shape over sequence readout are beyond the scope of this study, and for this reason we prefer not to provide a full extended table such as this one in the Supplementary Material.

Minor comments:

1. Figure 1, R2 values overall are not high, even though they are consistently better when incorporating higher order variables. Are those low R2 values typical for these algorithms with selex data?

R: An analysis of HT-SELEX datasets done by [Yang *et al.* 2017] obtained average R^2 values of around 0.60 and 0.70 for *1mer* and *1mer+shape* models, respectively. In our case, the equivalent R^2 values obtained with CAP-SELEX data are lower (~ 0.24 and ~ 0.31 , respectively), but do show a consistent improvement. Two main issues that explain the reduction of overall R^2 values are (i) amplification biases in the CAP-SELEX datasets for non-TF related motifs, and (ii) low coverage and robustness for long k -mers.

We argue that these values are expected given the R^2 values obtained when comparing technical replicates of the CAP-SELEX datasets. To demonstrate that, we calculated the R^2 values from relative affinity values for 10-mers from technical replicates (selected by similarity to the reference k -mers). This distribution of R^2 -values (see blue density in **Reviewer Figure 6**) was then compared with the R^2 values obtained from our best models used to predict relative affinities using encoded k -mer features (see red density). From a subset of 134 datasets with technical replicates (same TF-TF pair, same CAP-SELEX cycle, different barcode), we used the k -mers most similar to the reference k -mers, based on our motif matching step. The R^2 values from the technical replicates are highly variable (IQR=0.49), with a median at 0.46 (lower than observed for the best models in HT-SELEX), whereas the R^2 values from the modeling show a Gaussian distribution around a median of 0.33 (IQR=0.19). Given the technical variability (according to (i) or (ii) in the previous paragraph), our R^2 -values from the modeling are expected to be lower than 0.46 (median).

The maximum test R^2 values obtained in our models are therefore globally lower than the ones reported for other SELEX setups (HT-SELEX), or even other technologies such as Protein Binding Microarrays, yet are reasonable and expected by the R^2 distribution for technical replicates.

Reviewer Figure 6. Coefficients of determination between technical replicates versus best models in CAP-SELEX data. Shown are the distributions of R^2 values for technical replicates (blue) and best models (red) for the prediction of 10 k -mers selected in the motif matching step using reference motifs reported in Jolma *et al.* 2015. Vertical blue and red lines indicate the median R^2 value in each category (values on top).

2. Line 143, change 'bi-specific Forkhead motif' to alternate forkhead motif or FHL motif

Thank you for the comment. We have adopted that expression in the revised manuscript

Updates in Text:

- (line 152) "alternate Forkhead motif (FHL)"

3. On p. 6, line 127, Fig S2c-d is incorrectly referred to, and should instead be Fig S1c-d. Also, I recommend this be moved to main body Fig 1.

R: We have fixed this incorrect reference and updated this panel in the main body Figures (1d).

Updates in Text/Figures:

- Fig S1c in the original version now has been moved to Fig 1c.

4. p. 12, bottom paragraph: how much apparent variability in degrees of cooperativity would you expect by chance alone? Permutation testing may be useful in assessing this.

R: Thank you for the comment. In our initial analysis we have observed 5, 24 and 18 enrichments that are considered significant for ω , ω -none or both sequences simultaneously between co-occupied and non-shared ChIP-seq peaks for Forkhead-Ets TFs (FDR = 10%).

In this revised version we have added a permutation scheme to assess the expected number of significant enrichments. We randomly subsample a number of Forkhead+Ets peaks equal to the initially observed number of co-occupied peaks and recalculate enrichment values for ω or ω -none motifs by matching and comparing those sequences between sampled and unsampled peaks. This has been done 100 times for paired TF datasets.

Additionally, we want to point out that we have found an error in our analysis that reduced the peak sets used for co-enrichment in certain cases, which we amended in this new version. This correction does not affect the conclusion of this section, but the overall numbers reported.

In this revised version, the number of significant enrichments observed were 36, 19 and 18, for ω -none, ω and both, respectively (by permutations: 25.1, 8.9 and 9.1; Z-scores = 2.9, 4.0, and 4.3) (Numbers in the previous version were 38, 20, and 15, respectively). We have updated those numbers in **Fig 4e**, added **Supp Material 6** to provide all those enrichments, and also updated the respective code in the Github.

Updated in Text/Figures:

We have

- updated **Fig 4e** to show the final enrichments reported, and the Venn diagram highlighting the consolidated enrichments.
- updated the following in the main text (**ln 320, underlined**): *“We found both ω -none and ω enriched among the co-occupied regions of the 126 TF pairs using the single occupied regions as background (**Fig 4e**). 18 TF pairs (expected by permutations = 8.9, Z-score = 4.0) showed enrichment for both the non-cooperative as well as cooperative sequences (ω -none and ω) confirming the co-existence of cooperative and non-cooperative binding patterns between the same pairs of TFs in vivo. Another 19 pairs (expected = 17.4; Z-score = 0.9) were only enriched for either cooperative or non-cooperative sequences (1 and 18 respectively).”*
- provided the enrichment values for TF pairs as a table in the **Supp Material 6**, refer to it in the main text (**line 320**) and added the respective code in our GitHub repository, to allow reproducibility.

5. p. 13, line 314: p-value is significance, but what is the effect size? Also, what is the ‘truth’ as a positive control in this analysis? Is higher p-val more accurate, or is it simply an effect of small #s?

R: We thank the reviewer for raising this point. In the initial version of this manuscript we have included ROC-AUC and PR-AUC values to compare model setups on the classification between positive and background terms. We think that those can be used to highlight differences between both distributions with a standardized metric.

Additionally, in this revised version we include both the effect sizes for each comparison and the N for each group. We have used the effect size formula $r = Z / \sqrt{N}$, where Z is the Wilcoxon test statistic. This formula is used as suggested by “Rosenthal et al 1994, *Parametric measures of effect size. In H. Cooper & L. V. Hedges (Eds.), The handbook of research synthesis. (pp. 231-244). New York: Russell Sage Foundation*”

We computed the effect sizes for the Wilcoxon rank sum test statistic, and included them in the main text next to reported P values. These values are ~ 0.3 (“TF1 and TF2” versus “background” terms for all model types), and ~ 0.1 (*peaks+k-mers* models versus *peaks* or *k-mers* only models, using “TF1 and TF2” ontologies only). These values are interpreted by the effect-size rule-of-thumb tables as “medium” (0.3) and “small” (0.1) effects, respectively.

Regarding the positive controls, our gold standard used for all analyses were always ontologies that contain both TFs (TF1 and TF2), or only one of those (TF1 or TF2) as listed genes (Fig 5a). The rationale of this is that such TFs would be regulating other genes of the ontology, and co-occupied peaks and k -mers in the vicinity of those genes would be able to predict such associations. In contrast to that, background terms or ontologies that contain neither TF1 or TF2 are assumed to not contain such a regulation, which could arguably be wrong due to missing data. The association probabilities obtained, however, allow the classification of positives and negatives better than random. We observe additional classification improvement when adding k -mer information, which overall supports the idea of TF-TF cooperativity can explain those TF-ontology relationships.

Regarding the comment on small numbers in **Fig 5b** we think this a reasonable contention. To discard an effect of small numbers in the statistical comparison of association probabilities, we have compared the same distributions using a sub-sampling approach to reduce the number of background terms to the number of “TF1 and TF2” terms ($N=2,817$). While the exact p -values are different, our observation in **Fig 5b** do not change: The statistical significance between “TF1 and TF2” versus “background” is stronger (i.e. lower P) than the one between models in the “TF1 and TF2” group. Thereby there are no effects due to small vs. big numbers in our analysis.

Updates in text/figures:

- The following effect size numbers have been included in the main text (**line 345, underlined**): “For all models, we observed higher association probabilities for ontology terms annotated to the TF pair (“TF1 and TF2”) than for random background terms (**effect size $r = 0.3$** , $P < 0.001$; Wilcoxon one-sided test) (Fig 5b; Methods). The highest associations were obtained for models considering genes regulated by cooperatively bound peaks and k -mers (**mean r between peaks+k-mers versus peaks or k-mers =**

0.1; mean $P < 0.01$), emphasizing the role of TF cooperativity in regulating specific processes.”

- We have included the numbers of elements for “TF1 and TF2” and “background” associations on the x-axis of **Fig 5b** (2,817 and 5,070,498).

6. p. 14, line 331-2: A description of the “model-dependent signal-to-noise threshold on the association probabilities” should be provided in the main text.

R: We thank the reviewer for this comment. We have included a sentence that summarizes the signal-to-noise threshold calculation in the main text.

Updates in text/figures:

- We have updated the following in the main text (**line 366**) “Briefly, we defined strong and weak associations between TF pairs and ontologies by shuffling the labels that describe whether a TF pair is part of the listed genes in the ontology (predicted variable) and recalculating the association probabilities that these decoy associations would generate when fitting a model to them (**Methods**)”.

7. No legend is provided for Fig S2c.

R: We thank the reviewer for pointing this out. In this updated version we provide a legend for **Fig S2c**, which is now re-labeled as **Fig S5d**.

Updates in text/figures:

- Added a legend of Figure S5 “(c) Full spectra for FOXO1:ETS1 interacting with ω -none. (d) Zoom views of insets in (c)”

8. Guturu et al., 2013 is cited but not listed in the references.

R: We have now included Guturu et al. 2013 in the references.

Updates in text/figures:

- (**line 549**): Guturu, H., Doxey, A. C., Wenger, A. M., & Bejerano, G. (2013). Structure-aided prediction of mammalian transcription factor complexes in conserved non-coding elements. *Philosophical Transactions of the Royal Society of London. Series B, Biological Sciences*, 368(1632), 20130029. <https://doi.org/10.1098/rstb.2013.0029>.

9. The way they're aligned in figure 2C, the forkhead protein would be in a different register with respect to the ETS protein for the ω -none and ω than with the ω -high. Are there spacing differences between motifs/most important kmers?

R: We thank the reviewer for raising this point. The most important k -mers in each CAP-SELEX dataset, as highlighted by Jolma *et al* (2015), can have differences in spacing between TFs. Multiple such spacings can be present in the same dataset, although with different prevalence. Despite their presence, one caveat in the TF-binding model analyses is that a known k -mer prior is needed, which is based on known TF motifs. In the case of FHL sites, it seems these motifs (i.e. GCACG) were not considered, as composite motifs with this pattern are not reported as enriched.

Regarding the spacing differences for Forkhead and Ets when co-binding to ω/ω -none and ω -high sites, we agree with the Reviewer's observation. An explanation for this is indicated in a previous comment (Reviewer comment #5).

10. The manuscript needs proofreading for proper English grammar.

R: We thank the reviewer for this suggestion and have asked a colleague to proofread the manuscript. Proofreading modifications are indicated in the main text with **magenta text**.

Reviewer #2 (Remarks to the Author):

1. In this densely written and data-rich manuscript, Ibarra and co-workers present a combined bioinformatic and experimental analysis of transcription factor (TF) cooperativity. Broadly speaking, they further analyzed the previously published CAP-SELEX database on human TF pairs (Jolma et al., Nature 527:384) to better identify DNA shape as a driver for cooperativity, and attempted to validate several aspects of their conclusions in vitro using NMR spectroscopy and isothermal titration calorimetry for ETS-Forkhead pairs and in vivo using phenotypes linked to TF associations. The manuscript is based extensively on bioinformatic approaches, for which I have little expertise, and thus it is hard for me to judge how much of an advancement it represents over past publications. Nevertheless, it remains a challenge to understand how TFs work in a cellular context to regulate gene expression, and this paper nicely highlights the complex role of DNA sequence in mediating TF cooperativity. This should be of interest to the readers of Nature Communications and I recommend that it be accepted for publication.

R: We thank the reviewer for the positive recommendation of this work.

2. With that stated, I do have expertise in the biophysical chemistry of DNA-protein interactions, and suggest that the authors consider the following hopefully helpful comments on their experimental result.

An intriguing conclusion of this paper lines with the statement on line 158 that the FOXO1-binding strength determined the level of cooperativity with ETS1. It would help if the authors clarified this statement in light of thermodynamic linkage. In such a coupled system, the free energy contribution for the FOXO1-ETS1-DNA cooperativity should add to the free energies of the individual protein-DNA interactions. Thus, the presence of FOXO1 should also increase the affinity of ETS1 for DNA (which unfortunately was not measured), and ETS1 cooperativity should increase the affinity of FOXO1 for both omega-none and omega-high sites. One possible explanation is that it is not the affinity of FOXO1 or ETS1 for a given site per se that matters. Rather, by changing DNA sequences, DNA-mediated cooperativity between ETS1 and FOXO1 may also have been changed. Separating such "direct" and "indirect" effects could be very difficult and may require studying a more extensive series of DNAs than the three listed in Fig 2c (which are somewhat biased having been selected from CAP-SELEX and HT-SELEX data sets to show or not show cooperativity).

We thank the reviewer for raising this point. We agree with the reviewer on the possibility of a more complex explanation for our experimental results. We have included in this revision a set of additional experiments for ETS1 that should increase the value of this work and better support the claims made.

Regarding the possibility of a more likely interaction between FOXO1 and ETS1, we want to emphasize that this complex is forming due to DNA-mediated interactions, rather than protein-

protein interactions. This is supported by the crystal structure of FOXO1:ETS1, which aligns well with the sequences ω , and ω -none used in our experiments. In the case of ω -high, the DNA-binding domains are predicted to be more separated (see Reviewer 1, comment #5), which further reduces the possibility of protein-protein interactions in this case. Of course, we cannot rule out that other parts of the proteins or even other cofactors might increase cooperativity in an *in vivo* setting, however, we do observe increased binding for some DNAs tested for FOXO1 in presence of ETS1, again confirming DNA shape mediated cooperativity.

ETS1 has similar, strong (nanomolar affinity) binding to the ω and ω -high DNA sequences tested experimentally, and also similar thermodynamic parameters (now shown as inset in **Fig S4**). Unfortunately, we cannot experimentally show that this would change when interacting with FOXO1, despite our attempts in a new round of experiments. Also, the ETS1 binding motif GGAAG site does not change between the three different DNAs tested, thus our results of unchanged ETS1 affinity is expected for the three sequences.

Unfortunately, the measurements of thermodynamic parameters for ETS1, added to an equimolar mixture of FOXO1-DNA, did not give interpretable results. A possible explanation might be that the concentration we are using in the cell is below the K_d of FOXO1 binding to this DNA. Now, if ETS1 is added to an equimolar ratio of FOXO1 and ω -high, multiple events could happen: ETS1 is going to bind the DNA first, and upon that binding the FOXO1 binding site on the DNA becomes accessible to FOXO1, even in low concentrations. If we would titrate the other way around, namely adding an equimolar ratio of FOXO1 and DNA from the syringe to ETS1 in the cell, this would be likewise problematic. The bound FOXO1-DNA complex in the syringe, where the concentration is high, gets diluted during the titration, meaning that FOXO1 might dissociate from the DNA. At the same time, ETS1 might bind to the DNA, followed by a new binding event of FOXO1 to this complex. Therefore, no matter which way we set up the experiment, we would never be sure, which binding events give rise to the measured heats.

In this revised version, we have included ITC experiments for ETS1 with ω -high and ω sequences and measured thermodynamic parameters from the titration curves, in replicates. Overall, our results do not support major changes in observed K_d and thermodynamic parameters between the two DNA sequences for ETS1. This suggests that the DNA mediated cooperativity during FOXO1 binding to ω -high is mainly affecting FOXO1 binding thermodynamics, and not ETS1 (see also comments #5 and #6).

Updates in Figures/Text:

We have:

- included ITC experiments for ETS1 binding to ω -high and ω sequences, in replicates. (**Fig S4b**).
- highlighted these measurements in the main text, in a new sentence (**lines 186**): “*The three sequences mainly differ in their FOXO1 binding region, while their ETS1 binding region contains a strong Ets binding site (Fig 2c)*”

- updated the following sentence (**line 194**): “As *ETS1* alone binds with similarly high affinities to ω and ω -high (ω -none could not be measured by ITC), we propose that this cooperativity is mediated by DNA, which is also supported by a crystal structure [Choy et al. 2014], showing no direct interactions between FOXO1 and ETS1 (**Fig 3e**).”

3. “Given the limited information in the Methods section and Supplemental Material 3, it is very difficult to judge how the ITC experiments were carried out. What exact DNA oligonucleotides were used, how were they prepared, and how were concentrations of the proteins and DNA determined? Was the *Ets1* included with DNA in the syringe (likely the case) or with the FOXO1 in the cell? What was the concentration of ETS1 and was the DNA saturated?...”

R: We thank the reviewer for raising this concern regarding the Methods on the ITC data generation and analysis and apologize for omitting experimental details in the first version. We agree with the reviewer that it can be difficult to interpret all details for the ITC experiments from the original Methods sections, so in this updated version we have provided an extensive description of the requested details, plus visualizations that should help in the interpretation of how these experiments were done.

Among the most important details, in this version we have: (i) described the exact sequences in the **Supplementary Material 3** and **Fig 2c** used for both ITC and NMR, (ii) described preparation details and (iii) concentrations for all experiments, (iv) clarified what has been added to the cell and syringe, using illustration panels next to the ITC curves, and (v) provide specifics on concentration and saturation points.

Updates in Text/Figures:

Regarding the numbered updates:

- i) We have indicated the three DNA sequences in the table **Supp Material 3** (sheet “DNA sequences”)
- ii) Preparation details are indicated in the Methods section (**line 895, underlined**): “**DNA oligonucleotide annealing:** Both oligo strands were added in equimolar ratios in a buffer containing 10 mM Tris pH 7.5, 50 mM NaCl and 1 mM EDTA. Afterwards, the samples were incubated for 5 min at 95°C and cooled down 0.1 degree per second. The concentration of the aligned DNA molecules was determined by measuring absorbance at 260 nm using a Nanodrop2000.”. Additionally, we have included a statement regarding protein concentrations and purity (**line 901**): “The concentration of both proteins was measured using Nanodrop2000 and their absorbance at 280 nm. The purity of both proteins was determined by SDS gel electrophoresis and the ratio of absorbance at 260 nm and 280 nm.”
- iii) In the same supplementary material indicated in (i), the concentrations used in the ITC experiments for both cell and syringe are highlighted in the sheet “ITC” (columns “[Syr] (M)” and “[Cell] (M)”).
- iv) We have described what the syringe and cell compositions were in the Methods (**line 914**): “While *ETS1* was kept in the cell and DNA was added from the syringe, FOXO1

was always titrated from the syringe into the cell filled with DNA or DNA and ETS1 in equimolar ratios.“ Additionally, we have updated illustration schemes for injection (**Fig S4 and Fig S8**).

v) Syringe and injection details are now updated in the Methods (**line 912**): *“The first injection was 0.4 μ l, and 2 μ l for the remaining nineteen. For the ETS1 titrations against the different DNAs 13 injections of 3 μ l were done. While ETS1 was kept in the cell and DNA was added from the syringe, FOXO1 was always titrated from the syringe into the cell filled with DNA or DNA and ETS1 in equimolar ratios.”*

4. *“Can the authors demonstrate that the measured heats were strictly from the FOXO1 binding the DNA or the saturated ETS1-DNA complex (and without contributions from changes in the ETS1-DNA interactions)?”*

R: This is indeed an interesting question, which we are unfortunately not able to answer entirely.

As mentioned in our ITC descriptions, ETS1 is pre-saturated in the cell with the DNA sequence ω , before addition of FOXO1. That should reduce the detection signal of ETS1-DNA binding events in our setup. Also, as mentioned already (**Reviewer 2 - comment #2**), we have performed ITC measurements of ETS1 and DNA (titrating DNA from the syringe into the cell, filled with ETS1). While the thermodynamic profiles for FOXO1 binding to DNA or the ETS1-DNA complex are very similar and mostly enthalpy driven, ETS1-DNA interactions exhibit large entropy changes (see **Reviewer Figure 7** and insets in **Fig S4**), which are not visible in FOXO1 titrations into ETS1-DNA. Thus, we think that the measured heats are mostly from FOXO1 binding to the saturated ETS1-DNA complex.

One caveat is the possibility that the increase of entropy is due to the DNA opening up after dilution from the syringe into the cell, an observation often made for structured RNAs, for example. This would make our ITC measurements for DNA to ETS1 useless. However, when we titrate DNA into buffer we do observe endothermic heat spikes (see **Reviewer Figure 8** and **Fig S4d**) but they are much weaker than in presence of ETS1. Furthermore, they do not change over the titration. As the heat spikes are larger in presence of ETS1 in the beginning and then get smaller until they reach the volume of those in DNA-buffer titration, we conclude we can indeed determine the ETS1-DNA affinity from these measurements and that they are similar for all DNAs.

Reviewer Figure 7. Thermodynamic parameters for FOXO1- and ETS1-DNA interactions measured by ITC. (left) Thermodynamic parameters values measures for FOXO1 added into a cell with DNA sequence ω (center), a cell with ω +ETS1, and (right) ω added to a cell with ETS1.

Reviewer Figure 8. Heat plot for DNA sequences injected into buffer measured by ITC. (top) Titration scheme. DNA (strands) is titrated from the syringe into a cell with buffer. (bottom) Heat peaks for DNA sequences ω -high and ω are shown.

Updates in Text/Figures:

- We have added a **Reviewer Figure 7** with thermodynamic values and control ITC titrations as an extended **Fig S4**.
- We have updated the reference to these supplementary figures in the main text (**line 188, underlined**): “For FOXO1 alone, we observed a 10-fold stronger binding for ω -none than for ω ($K_d = 24 \pm 3$ nM and 352 ± 22 nM, respectively; $P < 0.01$, two-sided t-test **Fig 2d; Fig S4a**).”

5. “Also, in Figure S2a, the raw ITC data seem to rather noisy with significant changes even for endpoints (presumably heats of dilution) which likely impacts the reliability of the fit data.

Furthermore, the raw data in the left panels of Figure S2a do not match the fit results give in the insets (likely reversed top/bottom). The reason I point this out is that it is extremely difficult to reliably measure K_d values for even simple protein-DNA interactions (reported literature results on the same systems often differ by orders of magnitude), and even more of a challenge to dissect cooperative interactions in multi-component systems. “

R: We acknowledge that the noise in the endpoints can be rather high and affect the fitting. To increase the confidence level of these results, we have included an additional replicate for all ITC measurements related to FOXO1. These trends can be reliably recovered in independent measurements. (see **Fig S4**). We also agree that it is difficult to make a definitive conclusion from our ITC and NMR measurements, especially regarding the unexplained difficulties in measuring some pairs using ITC (i.e. FOXO vs. ω -high and ETS1 vs ω -none), which is the reason why we also used NMR. But we think that we altogether can confirm a DNA-dependent modulation of affinities, predicted from the bioinformatic analysis in the main part of the manuscript. We therefore added a small paragraph to point out our difficulties and to tone down our conclusions.

Regarding the differences between inset values and raw data in the original manuscript version, we apologize for this mistake. We have corrected the panels and insets accordingly.

Updates in Text/Figures:

- 1) We provide all ITC plots for FOXO1, ETS1 and FOXO1:ETS1 WT (**Fig S7**, as indicated in **Reviewer 2 - comment #4**)
- 2) Updated the following statement in the main results (**line 210**): *“The fact that we cannot measure ITC for every possible pair (FOXO1- ω -high and ETS1- ω -none) shows that there are questions remaining about the biophysical nature of these interactions, which are difficult to answer in such a multi-component system. However, we could confirm the DNA-dependent affinity changes for FOXO1 and its modulation in presence of ETS1.”*

6. “It might also be difficult for most readers to follow the simplified and perhaps overly optimistic discussion of the NMR-monitored titrations (lines 179 onward). In contrast to what is stated, in Fig 2e, K192 of FOXO1 shows slow exchange behavior when titrated with both omega-none and omega, without or with DNA (G230 in Fig S2c might be an example of what the authors are trying to conclude). In the case of omega-high, K192 of FOXO1 shows fast exchange with omega-high and only a slight change in lineshape when ETS1 is present. It is not clear that this is a switch from fast to intermediate exchange and hence indicative of cooperative binding (especially since the consequences of exchange on NMR spectra are dependent upon chemical shift differences, kinetics, and relaxation, with the latter certainly differing due to the molecular weight difference between the FOXO1-DNA and FOXO1-DNA-ETS1 complexes).

R: We agree with the reviewer in that our original presentation of the results might come across as too simplistic, and therefore might not visually support a cooperativity interaction between FOXO1-ETS1 for ω -high. In the revised version we have extended the presentation of our

results to support the main claim more thoroughly (see **Fig S5 and S6**). The three figures together (**Fig S5 and S6** and **Fig 2e**) should now better show the change in lineshape between absence and presence of ETS1 upon ω -high titration as opposed to ω -none titration. The resulting binding curves in **Fig 2e** show for all used residue a change from a fast-exchange and fittable curve to a more sigmoidal non-fittable curve due to signal loss and intermediate titration steps. That peaks do not reappear as should be the case for slow exchange (like in **Fig S6 a/b**) for ω -high in presence of ETS1 is due to the increased size of the complex, which further leads to line broadening (also observable in **Fig S6 c/d**). This is also brought up by the reviewer and we could compensate for this increase of linewidth by using TROSY-type experiments. However, this would simply lead to having more beautiful spectra and does not justify the additional time and costs for deuteration, etc. We believe that the spectra show a clear change of chemical shift perturbations (regarding linewidth and exchange regime) and thus confirm the ITC data further with regards to ω -high and also confirm the already strong binding of FOXO1 to ω -none.

Among the main changes in the revised manuscript, we have changed **Fig 2e** to display more titration curves for the most relevant (with regards to chemical shift perturbation change from fast to intermediate/slow) residues, highlighting that this change is observed not just for K192, but for several peaks. Additionally, we have included more NMR spectra in **Fig S5** and **S6**.

Thus, we think that the conclusions drawn from our interpretations of NMR and ITC data remain the same (see our additions mentioned above).

Updates in Text/Figures:

- We have updated **Fig 2e** to show eight titration peaks for residues of FOXO1 upon interaction with ω -high. Regarding G230, we want to mention that this peak was not listed in the final selection of residues for visualization, due to dynamic range considerations.
- We have included two supplementary figures, showing FOXO1-DNA titration peaks, with and without ETS1, for ω -high and ω -none DNA sequences (**Fig S5 and S6**). We have updated the referral of those in the main text (**line 198, underlined**): *“Because we were unable to measure ω -high and FOXO1 using ITC, we resorted to measuring NMR chemical shift perturbations (**Fig 2e; S5**), interpreted as weak, moderate, or strong binding depending on the exchange regime (fast, medium, slow) to assess cooperativity between FOXO1 and ETS1. The results for ω -high were corroborated qualitatively by NMR, as chemical shift perturbations switched from fast- to intermediate-exchange regimes, indicating an increase in binding affinity for ω -high in presence of ETS1 (**Fig 2e, S5**). Additional peaks also show a similar behaviour (**Fig S2c**). For ω -none, we observed slow-exchange (stronger binding) for both FOXO1 alone and in the presence of ETS1 (**Fig S6**).”*
- We have removed the following sentence from the revised manuscript, as multiple peaks are now mentioned in **Fig 2e (line 204)**: *“Additional peaks also show a similar behaviour (Fig S2c).”*

7. *“Also, in Fig. 2f, the isotherms seems rather similar for FOXO1 binding omega-high DNA without or with ETS1. “*

R: We disagree with this interpretation of the isotherm curve. In our view this is a very different curvature. Understanding that this could be a matter of visual and mixed interpretation, we have included additional isotherm curves for several residues in the revised version of the manuscript. They all show the change from a typical fast exchange binding curve to a more sigmoidal curve and to line broadening (intermediate exchange) of intermediate titration points and are not possible to fit. This is also evident from the actual NMR spectra now fully provided in **Fig S5** and **S6**.

Updates in Text/Figures:

- As indicated in (**Reviewer 2 - comment #6**), we have included multiple isotherms curves that support the claim of cooperativity (see **Fig 2f**).

8. *“Taken together, it is unclear that there is any solid experimental evidence for ETS1-FOXO1 cooperativity with omega-high DNA”*

R: We also disagree with this point because of the reasons mentioned above (multiple isotherms showing changes between FOXO1 and FOXO1+ETS1, highlighted in the (**Reviewer 2 - comment #6**), and ITC data. Again, we do not claim that there is direct cooperativity between ETS1 and FOXO1 (or direct interaction) but rather that the increase in affinity by one order of magnitude from the absence to the presence of ETS1 for ω -high is indicative of DNA-mediated cooperativity. Or rather that the binding of ETS1 to DNA allows FOXO1 to bind better, most likely through a conformational change in DNA triggered by the interaction of ETS1 with the minor groove. This is only valid for ω -high. Thus, cooperative binding is DNA sequence/shape dependent.

9. *“Similar to the ITC methods, it is also difficult to judge how the NMR experiments were carried out and interpreted. For example, Supplemental Material 3 lists G161 as one NMR-monitored signal, yet data is not shown and it is not identified in Fig S2b. G230 is also tabulated to have measurable chemical shift changes, yet the peaks shown in the spectra of Fig S2c seem unchanged over the titration series.”*

R: We agree that judging on how the NMR experiments were done is difficult with the current NMR results and Supplementary Material. In this revised version we therefore have included an additional, and more comprehensive results description for NMR analyses. We have also included titrated data values for all peaks highlighted in this work (as chemical shift perturbation values).

Updates in Text/Figures:

We have:

- added all the results for titrated and fitted CSP values in **Supp Material 3**, sheet name “NMR”
- updated a formula in Methods describing the CSP calculation (**line 935**): “Chemical shift perturbations were calculated with the following formula: $CSP = \sqrt{\Delta H^2 + (0.2\Delta N)^2}$ ”. Additionally, we have included a reference for this formula [**Williamson et al 2013**]

10. *“Furthermore, K192 is tabulated to have chemical shift changes of 0.127 ppm and 0.223 ppm upon addition of 120 uM omega-high DNA without and with ETS1 present, respectively, yet in the lower panels of Fig 2e, very similar changes of ca. 0.12 ppm in the 1H dimension are seen.”*

R: We acknowledge this disagreement between the figure and the tabulated supplementary table values. This has been due to an unfortunate data-transfer error. In the revised version, we have made sure to update the values and to check that all columns in the spreadsheet are correct.

Updates in Text/Figures:

- The CSP values for K192 and its fitted representation are in the **Supp Material 3**, sheet “NMR”, columns **H** (“K192”) and **I** (“K192 fit”), respectively. Other residues are tabulated in this sheet as well

11. *“Finally, it is hard to be convinced by the authors' conclusion that Arg409 is the “driver” of FOXO1/ETS1 cooperativity. As noted above in the comments on thermodynamic cooperativity, it seems possible that the results of Fig 3f could simply reflect a reduction in the affinity of ETS1 for DNA due to the mutation, which then manifests as an apparently smaller increase in the affinity of FOXO1 for omega-medium DNA.”*

R: We understand the point raised by the reviewer. Indeed, we have tried to bring across the point that other residues may contribute to the cooperativity as well (**lines 291-293** in the revised version). In the revised version we tried to make this more clear and have rephrased certain things to not over-emphasize our previous statements. Specifically, we tuned down the usage of “driver” in the main section of our results.

As for the second point, we performed an additional ITC-experiment, to measure K_d values for the mutant ETS1 (R409A) to DNA. The affinity does not change significantly compared to wild type (**Reviewer Figure 9**). This result supports that the binding affinity of ETS1 is not reduced in the mutant, and that the R409A mutation has mostly an effect on FOXO1 binding.

Reviewer Figure 9. Dissociation constants of ETS1- ω complexes. Kd values for ETS1 WT (two replicates) and mutant R409A (one replicate)

Updates in Text/Figures:

- The Kd values for ETS1 WT and mutant are included in the **Supp Material 3** for ITC (sheet “ITC”)
- **(lines 284)** *“To exclude that the reduction in FOXO1 affinity to ω is solely driven by a decrease in ETS1 affinity upon the R409A mutation, we performed ITC measurement of ω into ETS1-R409A, and observed no significant changes in affinity compared to ETS1 wild type (Fig S8a-b).”*
- **(lines 288)** *“We concluded from these analyses that the DNA-mediated cooperativity between FOXO1 and ETS1 is strongly linked to the interaction of R409 of ETS1 and the DNA minor groove opposite to the FOXO1 binding site”.*
- New title section **(line 264)**: *“Site-directed mutagenesis reveals the role of Ets residue R409 in Forkhead-Ets cooperativity.”*

12. *“The actual experimental conditions (including the sequence of omega-medium) and ITC data for this section of the paper have not been presented.”*

R: We apologize for this confusion, ω -medium is in fact ω (we relabeled it during the writing process and forgot to change several instances in the supplementary figures).

The updates for ITC data are extensively described in a previous comments (**Reviewer 2 - comment #3**), where we highlight the updates for these sections (NMR+ITC). We thank the reviewer for highlighting this point.

Updates in Text/Figures:

- The label “ ω -medium” have been replaced with “ ω ” (three instances, in the legends of **Fig 3** and **Fig S9**, and in the title of **Fig S10f**)

13. *“Also, this highlights the need for additional experimental results, such as measuring the Kd values of WT and mutant ETS1 for DNA in the absence and presence of FOXO1”*

R: As indicated in (**Reviewer 2 - comment #11**), we have generated ITC data for ETS1 mutant and WT. We cannot measure the Kd for ETS1 in presence of FOXO1, both for the WT and mutant as mentioned earlier. (**Reviewer 2 - Comment #2**)

14. *“Although obviously well beyond the scope of this paper, it would also be interesting to compare the crystal structures of the FOXO1-ETS1-DNA complex with containing WT and A409A ETS1. In particular, if the author's conclusion regarding Arg409 holds true, then the conformation of the DNA should differ.”*

R: We agree with this suggestion from the reviewer as an interesting follow-up, but it is indeed beyond the scope of this manuscript. Reviewer 1 has requested additional computational analyses that go into a similar direction, namely a comparison of DNA conformations of ω -none/ ω / ω -high sequences. For that, we looked into the crystal structure of FOXO1:ETS1:DNA (please refer to **Reviewer 1 - comment #6**). So far, the analysis was not conclusive regarding a potential DNA change upon addition of ETS1, but it suggests that the DNA minor groove width can differ between ω -none and ω / ω -high, being lower for the former sequences in the vicinity of Arg409.

REVIEWERS' COMMENTS:

Reviewer #1 (Remarks to the Author):

The authors have addressed the prior major concerns.

Re. Reviewer Fig 5 in the rebuttal which is now Figure 2b in the revised manuscript: it's not clear what the authors mean by "motif matches". Do they actually mean positions matching the consensus motif? This needs to be clarified in the manuscript.

Reviewer #2 (Remarks to the Author):

Based on an extensive list of recommended revisions by two referees, the authors have substantially improved their manuscript. These improvements come both in the form of new experimental data and clarified explanations of the original data. Although some points remain open to debate and hence potential future studies, I recommend that the revised manuscript be accepted for publication.

Response to reviewers.

Reviewer #1 (Remarks to the Author):

The authors have addressed the prior major concerns.

Re. Reviewer Fig 5 in the rebuttal which is now Figure 2b in the revised manuscript: it's not clear what the authors mean by "motif matches". Do they actually mean positions matching the consensus motif? This needs to be clarified in the manuscript.

We thank the reviewer for highlighting this point. We have updated the motif matches definition with a parenthesis update in the Figure Legend, which is in the new figure legend, to clearly define what we mean.

Updated in main text:

- "(number of matches between Forkhead or Ets consensus k-mer and CAP-SELEX k-mer in assessed region)" (**Figure Legend 2c**).

Updates in Figures:

- The correlation value in figure 2c (bottom-right) is indeed positive, and the reported number in the reviewer manuscript was written down with a negative symbol. This is a typo, so we have modified that in the main figure panel 2c (i.e. from -0.60 to 0.60)

Reviewer #2 (Remarks to the Author):

Based on an extensive list of recommended revisions by two referees, the authors have substantially improved their manuscript. These improvements come both in the form of new experimental data and clarified explanations of the original data. Although some points remain open to debate and hence potential future studies, I recommend that the revised manuscript be accepted for publication.

We thank the reviewer for acknowledging the manuscript improvement, and the recommendation of acceptance.